# On the Safety of Interpretable Machine Learning: A Maximum Deviation Approach

**Dennis Wei**
IBM Research
dwei@us.ibm.com

**Rahul Nair**
IBM Research
rahul.nair@ie.ibm.com

**Amit Dhurandhar**
IBM Research
adhuran@us.ibm.com

**Kush R. Varshney**
IBM Research
krvarshn@us.ibm.com

**Elizabeth M. Daly**
IBM Research
elizabeth.daly@ie.ibm.com

**Moninder Singh**
IBM Research
moninder@us.ibm.com

## Abstract

Interpretable and explainable machine learning has seen a recent surge of interest. We focus on safety as a key motivation behind the surge and make the relationship between interpretability and safety more quantitative. Toward assessing safety, we introduce the concept of *maximum deviation* via an optimization problem to find the largest deviation of a supervised learning model from a reference model regarded as safe. We then show how interpretability facilitates this safety assessment. For models including decision trees, generalized linear and additive models, the maximum deviation can be computed exactly and efficiently. For tree ensembles, which are not regarded as interpretable, discrete optimization techniques can still provide informative bounds. For a broader class of piecewise Lipschitz functions, we leverage the multi-armed bandit literature to show that interpretability produces tighter (regret) bounds on the maximum deviation. We present case studies, including one on mortgage approval, to illustrate our methods and the insights about models that may be obtained from deviation maximization.

## 1   Introduction

Interpretable and explainable machine learning (ML) has seen a recent surge of interest because it is viewed as a key pillar in making models trustworthy, with implications on fairness, reliability, and safety [1]. In this paper, we focus on *safety* as a key reason behind the demand for explainability. The motivation of safety has been discussed at a qualitative level by several authors [2–5]. Its role is perhaps clearest in the dichotomy between directly interpretable models vs. post hoc explanations of black-box models. The former have been called "inherently safe" [3] and promoted as the only alternative in high-risk applications [5]. The crux of this argument is that post hoc explanations leave a gap between the explanation and the model producing predictions. Thus, unusual data points may appear to be harmless based on the explanation, but truly cause havoc. We aim to go beyond these qualitative arguments and address the following questions quantitatively: 1) What does safety mean for such models, and 2) how exactly does interpretability aid safety?

Towards answering the first question, we make a conceptual contribution in the form of an optimization problem, intended as a tool for assessing the safety of supervised learning (i.e. predictive) models. Viewing these models as functions mapping an input space to an output space, a key way in which these models can cause harm is through grossly unexpected outputs, corresponding to inputs that are poorly represented in training data. Accordingly, we approach safety assessment for a model by determining where it deviates the most from the output of a *reference model* and by how much (i.e., its *maximum deviation*). The reference model, which represents expected behavior and is deemed to be

36th Conference on Neural Information Processing Systems (NeurIPS 2022).

safe, could be a model well-understood by domain experts or one that has been extensively "tried and tested." The maximization is done over a *certification set*, a large subset of the input space intended to cover all conceivable inputs to the model. These concepts are discussed further in Section 2.

Towards answering the second question, in Section 4 we discuss computation of the maximum deviation for different model classes and show how this is facilitated by interpretability. For model classes regarded as interpretable, including trees, generalized linear and additive models, the maximum deviation can be computed exactly and efficiently by exploiting the model structure. For tree ensembles, which are not regarded as interpretable, discrete optimization techniques can exploit their composition in terms of trees to provide anytime bounds on the maximum deviation. The case of trees is also generalized in a different direction to a broader class of piecewise Lipschitz functions, which we argue cover many popular interpretable functions. Here we show that the benefit of interpretability is significantly tighter regret bounds on the maximum deviation compared with general black-box functions, leveraging results from the multi-armed bandit literature. More broadly, the development of tailored methods for additional model classes is beyond the scope of this first work on the maximum deviation approach (the black-box optimization of Section 4.4 is applicable to all models but obviously not tailored). We discuss in Appendix B.5 some possible approaches, and the research gaps to be overcome, for neural networks and to make use of post hoc explanations, which approximate a model locally [6–8] or globally [9, 10].

In Section 5, we present case studies that illustrate the deviation maximization methods in Section 4 for decision trees, linear and additive models, and tree ensembles. It is seen that deviation maximization provides insights about models through studying the feature combinations that lead to extreme outputs. These insights can in turn direct further investigation and invite domain expert input. We also quantify how the maximum deviation depends on model complexity and the size of the certification set. For tree ensembles, we find that the obtained upper bounds on the maximum deviation are informative, showing that the maximum deviation does not increase with the number of trees in the ensemble.

Overall, our discussion provides a more quantitative basis for safety assessment of predictive models and for preferring more interpretable models due to the greater ease of performing this assessment.

## 2 Assessing Safety Through Maximum Deviation

We are given a supervised learning model $f$, which is a function mapping an input feature space $\mathcal{X}$ to an output space $\mathcal{Y}$. We wish to assess the safety of this model by finding its largest deviation from a given reference model $f_0 : \mathcal{X} \mapsto \mathcal{Y}$ representing expected behavior. To do this, we additionally require 1) a measure of deviation $D : \mathcal{Y} \times \mathcal{Y} \mapsto \mathbb{R}_+$, where $\mathbb{R}_+$ is the set of non-negative reals, and 2) a certification set $\mathcal{C} \subseteq \mathcal{X}$ over which the deviation is maximized. Then the problem to be solved is

$$\max_{x \in \mathcal{C}} D(f(x), f_0(x)). \tag{1}$$

The deviation is worst-case because the maximization is over all $x \in \mathcal{C}$; further implications of this are discussed in Appendix C. Note that (1) is different than typical robust *training* where the focus is to learn a model that minimizes some worst case loss, as opposed to finding regions in $\mathcal{X}$ where two already trained models differ significantly.

We view problem (1) as only a *means* toward the goal of evaluating safety. In particular, a large deviation value is not necessarily indicative of a safety risk, as two models may differ significantly for valid reasons. For example, one model may capture a useful pattern that the other does not. We thus think that it would be overly simplistic to regard the maximum deviation as just another metric to be optimized in selecting models. What large deviation values do indicate, however, is a (possibly) sufficient reason for further investigation. Hence, the maximizing solutions in (1) (i.e., the $\arg\max$) are of as much operational interest as the maximum values (this will be illustrated in Section 5).

We now elaborate on elements in problem formulation (1).

**Output space $\mathcal{Y}$.** In the case of regression, $\mathcal{Y}$ is the set of reals $\mathbb{R}$ or an interval thereof. In the case of binary classification, while $\mathcal{Y}$ could be $\{0, 1\}$ or $\{-1, +1\}$, these limit the possible deviations to binary values as well ("same" or "different"). Thus to provide more informative results, we take $\mathcal{Y}$ to be the space of real-valued scores that are thresholded to produce a binary label. For example, $y$ could be a predicted probability in $[0, 1]$ or a log-odds ratio in $\mathbb{R}$. Similarly for multi-class classification with $M$ classes, $\mathcal{Y} \subset \mathbb{R}^M$ could be a $M$-dimensional space of real-valued scores. In Appendix A, we discuss considerations in choosing the deviation function $D$ as well as models that abstain.

**Reference model $f_0$.** The premise of the reference model is that it should capture expected behavior while being "safe". The simplest case is for $f_0$ to be a constant function representing a baseline value, for example zero or a mean prediction. We consider the more general case where $f_0$ may vary with $x$. Below we give several examples of reference models to address the natural question of how they might be obtained. The examples can be categorized as 1) existing domain-specific models, 2) interpretable ML models validated by domain knowledge, and 3) extensively tested and deployed models. The first two categories are prevalent in high-stakes domains where interpretability is critical.

1. **Existing domain-specific models:** These models originate from an application domain and may not be based on ML at all. For example in consumer finance, several industry-standard models compute credit scores from a consumer's credit information (the FICO score is the best-known in the US). Similarly in medicine, scoring systems (sparse linear models with small integer coefficients) abound for assessing various risks (the $CHADS_2$ score for stroke risk is well-known, see the "Scoring Systems: Applications and Prior Art" section of [11] for a list of others). These models have been used for decades by thousands of practitioners so they are well understood. They may very well be improved upon by a more ML-based model, but for such a model to gain acceptance with domain experts, any large deviations from existing models need to be examined and understood.

2. **Interpretable models validated by domain knowledge:** Here, an interpretable ML model is learned from data and is validated by domain experts in some way, for example by selecting important input features or by carefully inspecting the trained model. We provide two real examples: In semiconductor manufacturing, process engineers typically want decision trees [12] to model their respective manufacturing process (e.g. etching, polishing, rapid thermal processing, etc.) since they are comfortable understanding and explaining them to their superiors, which is critical especially when things go out-of-spec. Hence, a tree built from data (or any model in general) would only be allowed to make automated measurement predictions if the features it highlights (viz. pressures, gas flows, temperatures) make sense for the specific process. Similarly, in predicting failures of industrial assets such as wind turbines, some failure data is available to train models but experts in these systems (e.g. engineers) may also be consulted. They have knowledge that can help validate the model, for example which components are most likely to cause failures or which environmental variables (e.g. temperature) are most influential.

3. **Extensively tested and deployed models:** A reference model may also be one that is not necessarily informed by domain knowledge but has been extensively tested, deployed, and/or approved by a regulator. For medical devices that use ML models, the US Food and Drug Administration (FDA) has instituted a risk-based regulatory system. Any system updates or changes, for instance changes in model architecture, retraining based on new data, or changes in intended use (e.g. use for pediatric cases for devices approved only for adults), need to either seek new approvals or demonstrate "substantial equivalence" by providing supporting evidence that the revised model is similar to a previously approved device. In the latter case, the reference model is the approved device and small maximum deviation serves as evidence of equivalence. As another example, consider a ML-based recommendation model for products of an online retailer or articles on a social network, where because of the scale, a tree ensemble may be used for its fast inference time as well as its modeling flexibility [13]. In this case, a model that has been deployed for some time could be the reference model, since it has been extensively tested during this time even though human validation of it may be limited. When a new version of the model is trained on newer data or improved in some fashion, finding its maximum deviation from the reference model can serve as one safety check before deploying it in place of the reference model.

**Certification set $\mathcal{C}$.** The premise of the certification set is that it contains all inputs that the model might conceivably be exposed to. This may include inputs that are highly improbable but not physically or logically impossible (for example, a severely hypothermic human body temperature of 27°C). Thus, while $\mathcal{C}$ might be based on the support set of a probability distribution or data sample, it does not depend on the likelihood of points within the support. The set $\mathcal{C}$ may also be a strict superset of the training data domain. For example, a model may have been trained on data for males, and we would now like to determine its worst-case behavior on an unseen population of females.

For tabular or lower-dimensional data, $\mathcal{C}$ might be the entire input space $\mathcal{X}$. For non-tabular or higher-dimensional data, the choice $\mathcal{C} = \mathcal{X}$ may be too unrepresentative because the manifold of

realistic inputs is lower in dimension. In this case, if we have a dataset $\{x_i\}_{i=1}^n$, one possibility is to use a union of $\ell_p$ balls centered at $x_i$,

$$\mathcal{C} = \bigcup_{i=1}^{n} \mathcal{B}_r^p[x_i], \qquad \mathcal{B}_r^p[x_i] = \{x \in \mathcal{X} : \|x - x_i\|_p \le r\}. \tag{2}$$

The set $\mathcal{C}$ is thus comprised of points somewhat close to the $n$ observed examples $x_i$, but the radius $r$ does not have to be "small".

In addition to determining the maximum deviation over the entire set $\mathcal{C}$, maximum deviations over subsets of $\mathcal{C}$ (e.g., different age groups) may also be of interest. For example, Appendix D.3 shows deviation values separately for leaves of a decision tree, which partition the input space.

## 3 Related Work

In previous work on safety and interpretability in ML, the authors of [3, 14] give qualitative accounts suggesting that directly interpretable models are an inherently safe design because humans can inspect them to find spurious elements; in this paper, we attempt to make those qualitative suggestions more quantitative and automate some of the human inspection. Furthermore, several other authors have highlighted safety as a goal for interpretability [2, 4, 15, 16, 5], but again without quantitative development. Moreover, the lack of consensus on how to measure interpretability motivates the relationship that we explore between interpretability and the ease of evaluating safety.

In the area of ML verification, robustness certification methods aim to provide guarantees that the classification remains constant within a radius $\epsilon$ of an input point, while output reachability is concerned with characterizing the set of outputs corresponding to a region of inputs [17]. A major difference in our work is that we consider *two* models, a model $f$ to be assessed and a reference $f_0$, whereas the above notions of robustness and reachability involve a single model. Another important difference is that our focus is *global*, over a comprehensive set $\mathcal{C}$, rather than local to small neighborhoods around input points; a local focus is especially true of neural network verification [18–27]. We also study the role of interpretability in safety verification. Works in robust optimization applied to ML minimize the worst-case probability of error, but this worst case is over parameters of $f$ rather than values of $x$ [28]. Thomas et al. [29] present a framework where during model training, a set of safety tests is specified by the model designer in order to accept or reject the possible solution.

We build on related literature on robustness and explainability that deals specifically with tree ensembles. Mixed-integer programming (MIP) and discrete optimization have been proposed to find the smallest input perturbation to 'evade' a classifier [30] and to obtain counterfactual explanations [31]. MIP approaches are computationally intensive however. To address this Chen et al. [32] introduce graph based approaches for verification on trees. Their central idea, which we use, is to discretize verification computations onto a graph constructed from the way leaves intersect. The verification problem is transformed to finding all maximum cliques. Devos et al. [33] expand on this idea by providing anytime bounds by probing unexplored nodes.

Safety has become a critical issue in reinforcement learning (RL) with multiple works focusing on making RL policies safe [34–37]. There are two broad themes [38]: (i) a safe and verifiable policy is learned at the outset by enforcing certain constraints, and (ii) post hoc methods are used to identify bad regimes or failure points of an existing policy. Our current proposal is complementary to these works as we focus on the supervised learning setup viewed from the lens of interpretability. Nonetheless, ramifications of our work in the RL context are briefly discussed in Appendix C.

## 4 Deviation Maximization for Specific Model Classes

In this section, we discuss approaches to computing the maximum deviation (1) for $f$ belonging to various model classes. We show the benefit of interpretable model structure in different guises. Exact and efficient computation is possible for decision trees, and generalized linear and additive models in Sections 4.1 and 4.2. In Section 4.3, the composition of tree ensembles in terms of trees allows discrete optimization methods to provide anytime bounds. For a general class of piecewise Lipschitz functions in Section 4.4, the application of multi-arm bandit results yields tighter regret bounds on the maximum deviation. While some of the results in this section may be less surprising, one of our

contributions is to identify precise properties that allow them to hold. We also show that intuitive measures of model complexity, such as the number of leaves or pieces or smoothness of functions, have an additional interpretation in terms of the complexity of maximizing deviation. More broadly, the development of methods specific to additional model classes is beyond the scope of a single work. We discuss in Appendix B.5 possible approaches and the advances needed for neural networks (beyond applying the black-box methods of Section 4.4) and to make use of post hoc explanations.

To develop mathematical results and efficient algorithms, we will sometimes assume that the reference model $f_0$ is from the same class as $f$. In Appendix C, we discuss the case where $f_0$ may not be globally interpretable, but may be so in local regions. We will also sometimes assume that the certification set $\mathcal{C}$ and other sets are Cartesian products. This means that $\mathcal{C} = \prod_{j=1}^{d} \mathcal{C}_j$, where for a continuous feature $j$, $\mathcal{C}_j = [\underline{X}_j, \overline{X}_j]$ is an interval, and for a categorical feature $j$, $\mathcal{C}_j$ is a set of categories. We mention relaxations of the Cartesian product assumption in Appendix B.2.

## 4.1 Trees

We begin with the case where $f$ and $f_0$ are both decision trees. A decision tree with $L$ leaves partitions the input space $\mathcal{X}$ into $L$ corresponding parts, which we also refer to as 'leaves'. We consider only non-oblique trees. In this case, each leaf is described by a conjunction of conditions on individual features and is therefore a Cartesian product as defined above. With $\mathcal{L}_l \subset \mathcal{X}$ denoting the $l$th leaf and $y_l \in \mathcal{Y}$ the output value assigned to it, tree $f$ is described by the function

$$f(x) = y_l \quad \text{if } x \in \mathcal{L}_l, \quad l = 1, \ldots, L, \tag{3}$$

and similarly for tree $f_0$ with leaves $\mathcal{L}_{0m}$ and outputs $y_{0m}$, $m = 1, \ldots, L_0$. As discussed in [39, 40], *rule lists* where each rule is a condition on a single feature are one-sided trees in the above sense.

The partitioning of $\mathcal{X}$ by decision trees and their piecewise-constant nature simplify the computation of the maximum deviation (1). Specifically, the maximization can be restricted to pairs of leaves $(l, m)$ for which the intersection $\mathcal{L}_l \cap \mathcal{L}_{0m} \cap \mathcal{C}$ is non-empty. The intersection of two leaves $\mathcal{L}_l \cap \mathcal{L}_{0m}$ is another Cartesian product, and we assume that it is tractable to determine whether $\mathcal{C}$ intersects a given Cartesian product (see examples in Appendix B.1).

For visual representation and later use in Section 4.3, it is useful to define a bipartite graph, with $L$ nodes representing the leaves $\mathcal{L}_l$ of $f$ on one side and $L_0$ nodes representing the leaves $\mathcal{L}_{0m}$ of $f_0$ on the other. Define the edge set $\mathcal{E} = \{(l, m) : \mathcal{L}_l \cap \mathcal{L}_{0m} \cap \mathcal{C} \neq \emptyset\}$; clearly $|\mathcal{E}| \leq L_0 L$. Then

$$\max_{x \in \mathcal{C}} D(f(x), f_0(x)) = \max_{(l,m) \in \mathcal{E}} D(y_l, y_{0m}). \tag{4}$$

We summarize the complexity of deviation maximization for decision trees as follows. This is a slight refinement of [32, Thm. 1] in the case $K = 2$, see Appendix B.1 for details.

**Proposition 1.** *Let $f$ and $f_0$ be decision trees as in* (3) *with $L$ and $L_0$ leaves respectively, and $\mathcal{E}$ be the bipartite edge set of leaf intersections defined above. Then the maximum deviation* (1) *can be computed with $|\mathcal{E}|$ evaluations as shown in* (4).

## 4.2 Linear and additive models

In this subsection, we assume that $f$ is a generalized additive model (GAM) given by

$$f(x) = g^{-1} \left( \sum_{j=1}^{d} f_j(x_j) \right), \tag{5}$$

where each $f_j$ is an arbitrary function of feature $x_j$. In the case where $f_j(x_j) = w_j x_j$ for all continuous features $x_j$, where $w_j$ is a real coefficient, (5) is a generalized linear model (GLM). We discuss the treatment of categorical features in Appendix B.2. The invertible link function $g : \mathbb{R} \mapsto \mathbb{R}$ is furthermore assumed to be monotonically increasing. This assumption is satisfied by common GAM link functions: identity, logit ($g(y) = \log(y/(1 - y))$), and logarithmic.

Equation (5) implies that $\mathcal{Y} \subset \mathbb{R}$ and the deviation $D(y, y_0)$ is a function of two scalars $y$ and $y_0$. For this scalar case, we make the following intuitively reasonable assumption throughout the subsection.

**Assumption 1.** For $y, y_0 \in \mathcal{Y} \subseteq \mathbb{R}$, 1) $D(y, y_0) = 0$ whenever $y = y_0$; 2) $D(y, y_0)$ is monotonically non-decreasing in $y$ for $y \geq y_0$ and non-increasing in $y$ for $y \leq y_0$.

Our approach is to exploit the additive form of (5) by reducing problem (1) to the optimization

$$M_\pm(f, \mathcal{S}) = \max/\min_{x \in \mathcal{S}} \sum_{j=1}^d f_j(x_j), \tag{6}$$

for different choices of $\mathcal{S} \subset \mathcal{X}$ and where $+$ corresponds to $\max$ and $-$ to $\min$. We discuss below how this can be done for two types of reference model $f_0$: decision tree (which includes the constant case $L_0 = 1$) and additive. For the first case, we prove the following result in Appendix B.2:

**Proposition 2.** *Let $f$ be a GAM as in (5) and $\mathcal{S}$ be a subset of $\mathcal{X}$ where $f_0(x) \equiv y_0$ is constant. Then if Assumption 1 holds,*

$$\max_{x \in \mathcal{S}} D(f(x), f_0(x)) = \max_{\sigma \in \{+, -\}} D\left(g^{-1}(M_\sigma(f, \mathcal{S})), y_0\right).$$

**Tree-structured $f_0$.** Since $f_0$ is piecewise constant over its leaves $\mathcal{L}_{0m}$, $m = 1, \ldots, L_0$, we take $\mathcal{S}$ to be the intersection of $\mathcal{C}$ with each $\mathcal{L}_{0m}$ in turn and apply Proposition 2. The overall maximum is then obtained as the maximum over the leaves,

$$\max_{x \in \mathcal{C}} D(f(x), f_0(x)) = \max_{m=1,\ldots,L_0} \max_{\sigma \in \{+, -\}} D\left(g^{-1}(M_\sigma(f, \mathcal{L}_{0m} \cap \mathcal{C})), y_{0m}\right). \tag{7}$$

This reduces (1) to solving $2L_0$ instances of (6).

**Additive $f_0$.** For this case, we make the additional assumption that the link function $g$ in (5) is the identity function, as well as Assumption 2 below. The implication of these assumptions is discussed in Appendix B.2.

**Assumption 2.** $D(y, y_0) = D(y - y_0)$ is a function only of the difference $y - y_0$.

Then $f_0(x) = \sum_{j=1}^d f_{0j}(x_j)$ and the difference $f(x) - f_0(x)$ is also additive. Using Assumptions 2, 1 and a similar argument as in the proof of Proposition 2, the maximum deviation is again obtained by maximizing and minimizing an additive function, resulting in two instances of (6) with $\mathcal{S} = \mathcal{C}$:

$$\max_{x \in \mathcal{C}} D(f(x), f_0(x)) = \max_{\sigma \in \{+, -\}} D\left(M_\sigma(f - f_0, \mathcal{C})\right).$$

**Computational complexity of (6).** For the case of nonlinear additive $f$, we additionally assume that $\mathcal{C}$ is a Cartesian product. It follows that $\mathcal{S} = \prod_{j=1}^d \mathcal{S}_j$ is a Cartesian product (see Appendix B.2 for the brief justification) and (6) separates into one-dimensional optimizations over $\mathcal{S}_j$,

$$\max/\min_{x \in \mathcal{S}} \sum_{j=1}^d f_j(x_j) = \sum_{j=1}^d \max/\min_{x_j \in \mathcal{S}_j} f_j(x_j). \tag{8}$$

The computational complexity of (8) is thus $\sum_{j=1}^d C_j$, where $C_j$ is the complexity of the $j$th one-dimensional optimization. We discuss different cases of $C_j$ in Appendix B.2; the important point is that the overall complexity is linear in $d$.

In the GLM case where $\sum_{j=1}^d f_j(x_j) = w^T x$, problem (6) is simpler and it is less important that $\mathcal{C}$ be a Cartesian product. In particular, if $\mathcal{C}$ is a convex set, so too is $\mathcal{S}$ (again see Appendix B.2 for justification). Hence (6) is a convex optimization problem.

### 4.3   Tree ensembles

We now extend the idea used for single decision trees in Section 4.1 to tree ensembles. This class covers several popular methods such as Random Forests and Gradient Boosted Trees. It can also cover *rule* ensembles [41, 42] as a special case, as explained in Appendix B.3. We assume $f$ is a tree ensemble consisting of $K$ trees and $f_0$ is a single decision tree. Let $\mathcal{L}_{l_k}$ denote the $l$th leaf of the $k$th tree in $f$ for $l = 1, \ldots, L_k$, and $\mathcal{L}_{0m}$ be the $m$th leaf $f_0$, for $m = 1, \ldots, L_0$. Correspondingly let $y_{l_k}$ and $y_{0m}$ denote the prediction values associated with each leaf.

Define a graph $\mathcal{G}(\mathcal{V}, \mathcal{E})$, where there is a vertex for each leaf in $f$ and $f_0$, i.e.

$$\mathcal{V} = \{l_k | \forall k = 1, \ldots, K, l = 1, \ldots, L_k\} \cup \{m | m = 1, \ldots, L_0\}. \tag{9}$$

Construct an edge for each overlapping pair of leaves in $\mathcal{V}$, i.e.

$$\mathcal{E} = \{(i, j) | \mathcal{L}_i \cap \mathcal{L}_j \neq \emptyset, \forall (i, j) \in V, i \neq j\}. \tag{10}$$

This graph is a $K + 1$-partite graph as leaves within an individual tree do not intersect and are an independent set. Denote $M$ to be the adjacency matrix of $\mathcal{G}$. Following Chen et al. [32], a maximum clique $S$ of size $K + 1$ on such a graph provides a discrete region in the feature space with a computable deviation. A clique is a subset of nodes all connected to each other; a maximum clique is one that cannot be expanded further by adding a node. The model predictions $y_c$ and $y_{0c}$ can be ensembled from leaves in $S$. Denote by $D(S)$ the deviation computed from the clique $S$. Maximizing over all such cliques solves (1). However, complete enumeration is expensive, so informative bounds, either using the merge procedure in Chen et al. [32] or the heuristic function in Devos et al. [33] can be used. We use the latter which exploits the $K + 1$-partite structure of $\mathcal{G}$.

Specifically, we adapt the anytime bounds of Devos et al. [33] as follows. At each step of the enumeration procedure, an intermediate clique $S$ contains selected leaves from trees in $[1, \ldots, k]$ and unexplored trees in $[k + 1, \ldots, K + 1]$. For each unexplored tree, we select a valid candidate leaf that maximizes deviation, i.e.

$$v_k = \underset{l_k, l_k \cap i \neq \emptyset, \forall i \in S}{\arg \max} D(S \cup l_k). \tag{11}$$

Using these worst-case leaves, a heuristic function

$$H(S) = D(S') = D(S \bigcup_{m=k+1}^{K+1} v_m) \tag{12}$$

provides an upper (dual) bound. In practice, this dual bound is tight and therefore very useful during the search procedure to prune the search space. Each $K + 1$ clique provides a primal bound, so the search can be terminated early before examining all trees if the dual bound is less than the primal bound. We adapt the search procedure of Mirghorbani and Krokhmal [43] to include the pruning arguments. Appendix B.3 presents the full algorithm. Starting with an empty clique, the procedure adds a single node from each tree to create an intermediate clique. If the size of the clique is $K + 1$ the primal bound is updated. Otherwise, the dual bound is computed. A node compatibility vector is used to keep track of all feasible additions. When the search is terminated at any step, the maximum deviation is bounded by $(D_{lb}, D_{ub})$.

The algorithm works for the entire feature space. When the certification set $\mathcal{C}$ is a union of balls as in (2), some additional considerations are needed. First, we can disregard leaves that do not intersect with $\mathcal{C}$ during the graph construction phase. A validation step to ensure that the leaves of a clique all intersect with the same ball in $\mathcal{C}$ is also needed.

## 4.4 Piecewise Lipschitz Functions

We saw the benefits of having specific (deterministic) interpretable functions as well as their extensions in the context of safety. Now consider a richer class of functions that may also be randomized with finite variance. In this case let $f$ and $f_0$ denote the mean values of the learned and reference functions respectively. We consider the case where each function is either interpretable or black box, where the latter implies that query access is the only realistic way of probing the model. This leads to three cases where either both functions are black box or interpretable, or one is black box. What we care about in all these cases[1] is to find the maximum (and minimum) of a function $\Delta(x) = f(x) - f_0(x)$. Let us consider finding only the maximum of $\Delta$ as the other case is symmetric. Given that $f$ and $f_0$ can be random functions $\Delta$ is also a random function and if $\Delta$ is black box a standard way to optimize it is either using Bayesian Optimization (BO) [44] or tree search type bandit methods [45, 46]. We repurpose some of the results from this latter literature in our context showcasing the benefit of interpretability from a safety standpoint. To do this we first define relevant terms.

---

[1]For simplicity assume $D(\cdot, \cdot)$ to be the identity function.

**Definition 1** (Simple Regret [45]). If $f_{\mathcal{C}}^*$ denotes the optimal value of the function $f$ on the certification set $\mathcal{C}$, then the simple regret $r_q^{\mathcal{C}}$ after querying the $f$ function $q$ times and obtaining a solution $x_q$ is given by, $r_q^{\mathcal{C}}(f) = f_{\mathcal{C}}^* - f(x_q)$.

**Definition 2** (Order $\beta$ c-Lipschitz). Given a (normalized) metric $\ell$ a function $f$ is c-Lipschitz continuous of order $\beta > 0$ if for any two inputs $x$, $y$ and for $c > 0$ we have, $|f(x) - f(y)| \leq c \cdot \ell(x, y)^{\beta}$.

**Definition 3** (Near optimality dimension [45]). If $\mathcal{N}(\mathcal{C}, \ell, \epsilon)$ is the maximum number of $\epsilon$ radius balls one can fit in $\mathcal{C}$ given the metric $\ell$ and $\mathcal{C}_{\epsilon} = \{x \in \mathcal{C} | f(x) \geq f_{\mathcal{C}}^* - \epsilon\}$, then for $c > 0$ the c-near optimality dimension is given by, $v = \max\left(\limsup_{\epsilon \to 0} \frac{\ln \mathcal{N}(\mathcal{C}_{c\epsilon}, \ell, \epsilon)}{\ln(\epsilon^{-1})}, 0\right)$.

Intuitively, simple regret measures the deviation between our current best and the optimal solution. The Lipschitz condition bounds the rate of change of the function. Near optimality dimension measures the set size for which the function has close to optimal values. The lower the value of $v$, the easier it is to find the optimum. We now define what it means to have an interpretable function.

**Assumption 3** (Characterizing an Interpretable Function). If a function $f$ is interpretable, then we can (easily) find $1 \leq m \ll n$ partitions $\{\mathcal{C}^{(1)}, ..., \mathcal{C}^{(m)}\}$ of the certification set $\mathcal{C}$ such that the function $f^{(i)} = \{f(x) | x \in \mathcal{C}^{(i)}\} \ \forall i \in \{1, ..., m\}$ in each partition is c-Lipschitz of order $\beta$.

*Note that the (interpretable) function overall does not have to be c-Lipschitz of bounded order, rather only in the partitions.* This assumption is motivated by observing different interpretable functions. For example, in the case of decision trees the $m$ partitions could be its leaves, where typically the function is a constant in each leaf ($c = 0$). For rule lists as well a fixed prediction is usually made by each rule. For a linear function one could consider the entire input space (i.e. $m = 1$), where for bounded slope $\alpha$ the function would also satisfy our assumption ($c = \alpha$ and $\beta = 1$). Examples of models that are not piecewise constant or globally Lipschitz are oblique decision trees (Murthy et al., 1994), regression trees with linear functions in the leaves, and functional trees. Moreover, $m$ is likely to be small so that the overall model is interpretable (viz. shallow trees or small rules). With the above definitions and Assumption 3 we now provide the simple regret for the function $\Delta$.

**1. Both black box models:** If both $f$ and $f_0$ are black box then it seems no gains could be made in estimating the maximum of $\Delta$ over standard results in bandit literature. Hence, using Hierarchical Optimistic Optimization (HOO) with assumptions such as $\mathcal{C}$ being compact and $\Delta$ being weakly Lipschitz [45] with near optimality dimension $v$ the simple regret after $q$ queries is:

$$r_q^{\mathcal{C}}(\Delta) \leq O\left(\left(\frac{\ln(q)}{q}\right)^{\frac{1}{v+2}}\right) \tag{13}$$

**2. Both interpretable models:** If both $f$ and $f_0$ are interpretable, then for each function based on Assumption 3 we can find $m_1$ and $m_0$ partitions of $\mathcal{C}$ respectively where the functions are $c_1$ and $c_0$-Lipschitz of order $\beta_1$ and $\beta_0$ respectively. Now if we take non-empty intersections of these partitions where we could have a maximum of $m_1 m_0$ partitions, the function $\Delta$ in these partitions would be $c = 2 \max(c_0, c_1)$-Lipschitz of order $\beta = \min(\beta_0, \beta_1)$ as stated next (proof in appendix).

**Proposition 3.** *If functions $h_0$ and $h_1$ are $c_0$ and $c_1$ Lipschitz of order $\beta_0$ and $\beta_1$ respectively, then the function $h = h_0 - h_1$ is c-Lipschtiz of order $\beta$, where $c = 2 \max(c_0, c_1)$ and $\beta = \min(\beta_0, \beta_1)$.*

Given that $\Delta$ is smooth in these partitions with underestimated smoothness of order $\beta$, the simple regret after $q_i$ queries in the $i^{\text{th}}$ partition $\mathcal{C}^{(i)}$ with near optimality dimension $v_i$ based on HOO is: $r_{q_i}^{\mathcal{C}^{(i)}}(\Delta) \leq O\left(q_i^{-1/(v_i+2)}\right)$, where $v_i \leq \frac{d}{\beta}$. If we divide the overall query budget $q$ across the $\pi \leq m_0 m_1$ non-empty partitions equally, then the bound will be scaled by $\pi^{1/(v_i+2)}$ when expressed as a function of $q$. Moreover, the regret for the entire $\mathcal{C}$ can then be bounded by the maximum regret across these partitions leading to

$$r_q^{\mathcal{C}}(\Delta) \leq O\left(\left(\frac{\pi}{q}\right)^{\frac{\beta}{d+2\beta}}\right) \tag{14}$$

Notice that for a model to be interpretable $m_0$ and $m_1$ are likely to be small (i.e. shallow trees or small rule lists or linear model where $m = 1$) leading to a "smallish" $\pi$ and $v$ can be much $>> \frac{d}{\beta}$ in case 1. Hence, interpretability reduces the regret in estimating the maximum deviation.

**3. Black box and interpretable model:** Making no further assumptions on the black box model and assuming $\Delta$ satisfies properties mentioned in case 1, the simple regret has the same behavior as (13). This is expected as the black box model could be highly non-smooth.

## 5    Case Studies

We present case studies to serve three purposes: 1) show that deviation maximization can lead to insights about models, 2) illustrate the maximization methods developed in Section 4, and 3) quantify the dependence of the maximum deviation on model complexity and certification set size (mostly in Appendix D). Two datasets are featured: a sample of US Home Mortgage Disclosure Act (HMDA) data (see Appendix D.2 for details), meant as a proxy for a mortgage approval scenario, and the UCI Adult Income dataset [47], a standard tabular dataset with mixed data types. A subset of results is shown in this section with full results, experimental details, and an additional Lending Club dataset in Appendix D. Since these are binary classification datasets, we take the deviation function $D$ to be the absolute difference between predicted probabilities of class 1. For the certification set $\mathcal{C}$, we consider a union of $\ell_\infty$ balls (2) centered at test set instances. While we have used the test set here, any not necessarily labelled dataset would suffice. The case $r = 0$ yields a finite set consisting only of the test set, while $r \to \infty$ corresponds to $\mathcal{C}$ being the entire domain $\mathcal{X}$. We reiterate that the dependence of the certification set on a chosen dataset is only on (an expanded version of) the support of the dataset and not on other aspects of the data distribution.

To demonstrate insights from deviation maximization, we study the solutions that maximize deviation (the $\arg\max$ in (1)) and discuss three examples below.

**Identification of an artifact:** The first example comes from the Adult Income dataset, where the reference model $f_0$ is a decision tree (DT) and $f$ is an Explainable Boosting Machine (EBM) [48], a type of GAM (plots of both in Appendix D.3). Here, the capital loss feature is the largest contributor to the maximum deviation (the discussion below Table 4 explains how this is determined), and Table 8 in Appendix D.3 shows that as the certification set radius $r$ increases, the maximizing values of capital loss converge to the interval $[1598, 1759]$. The plot of the GAM shape function $f_j$ for capital loss in Figure 1 shows that this interval corresponds to a curiously low value of the function. This low region may be an artifact warranting further investigation since it seems anomalous compared to the rest of the function, and since individuals who report capital losses on their income tax returns to offset capital gains usually have high income (hence high log-odds score). Note that this potential artifact was automatically identified through deviation maximization.

For the next two examples, we consider a simplified mortgage approval scenario using the HMDA dataset. Suppose that a DT $f_0$ (shown in Figure 3 in Appendix D.2) has been trained to make final decisions on mortgage applications. Domain experts have determined that this DT is sensible and safe and are now looking to improve upon it by exploring EBMs. (Logistic regression (LR) models are deferred to Appendix D.2 because they do not have higher balanced accuracy than $f_0$.)

**Conflict between $f, f_0$:** We first examine solutions that result in the most positive difference between the predicted probabilities of an EBM $f$ with parameter `max_bins=32` and the DT $f_0$. These all occur in a leaf of $f_0$ (leaf 2 in Figure 3) where the applicant's debt-to-income (DTI) ratio is too high ($> 52\%$) and $f_0$ predicts a low probability of approval. The other salient feature of the solutions is that they all have 'preapproval'=1, indicating that a preapproval was requested, which is given a large weight by the EBM $f$ in favor of approval (see Figure 4 in Appendix D.2, and Table 4 for more feature values). Thus $f$ and $f_0$ are in conflict. Among different ways in which the conflict could be resolved, a domain expert might decide that $f_0$ remains correct in rejecting risky applicants with high DTI, even if there is a preapproval request and the new EBM model puts a high weight on it.

**Trend toward extreme points, deviation can be good:** We now look at solutions that yield the most negative difference between the predicted probabilities of $f$ and $f_0$, for $r \le 0.8$. Table 1 shows the 6 features that contribute most to the deviation (again see Appendix D.2 for details). All of these points lie in a leaf of $f_0$ (leaf 14 in Figure 3, denoted $\mathcal{L}_{14}$) that excludes several clearer-cut cases, with the result being a less confident predicted probability of $0.652$ from $f_0$. The feature values that maximize deviation tend toward extreme points of the region $\mathcal{L}_{14}$. Specifically, the values of the continuous features debt-to-income ratio, loan-to-value ratio, property value, and income all move in the direction of application denial. For the latter three features, the boundary of $\mathcal{L}_{14}$ is reached as soon as $r = 0.1$, whereas for debt-to-income ratio, this occurs at $r = 0.4$. The movement

| $r$ | debt_to_income (%) | state | loan_to_value (%) | aus_1 | prop_value (000$) | income (000$) |
|-----|-----|-----|-----|-----|-----|-----|
| 0.0 | 46.0 | CA | 95.0 | 3 | 415 | 77.0 |
| 0.1 | [45.9 46.5] | none | [100. 100.92] | 1 | [120 120] | ≤28.5 |
| 0.2 | [45.5 46.5] | none | [100. 100.92] | 1 | [120 120] | ≤28.5 |
| 0.4 | [52. 52.] | none | [100. 100.92] | 1 | [120 120] | ≤28.5 |
| 0.6 | [52. 52.] | none | [100. 100.92] | 1 | [120 120] | ≤28.5 |
| 0.8 | [52. 52.] | none | [100. 100.92] | 1 | [120 120] | ≤28.5 |

Table 1: Values of top 6 features that maximize difference in predicted probabilities between a decision tree reference model $f_0$ and an Explainable Boosting Machine $f$ (`max_bins = 32`) on the HMDA dataset. For radius $r > 0$, the maximizing values of continuous features form an interval because the corresponding EBM shape functions $f_j$ are piecewise constant.

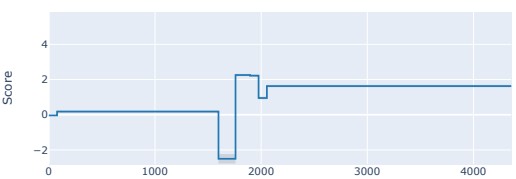

Figure 1: EBM shape function $f_j$ for capital loss feature, showing anomalously low interval identified by deviation maximization.

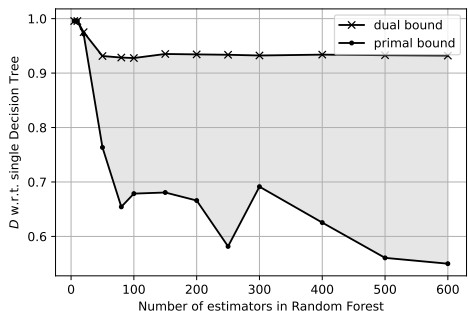

Figure 2: Maximum deviation $D$ on the Adult Income dataset as a function of number of estimators in a Random Forest.

toward extremes is expected for this $f$ since its relevant shape functions $f_j$ are mostly increasing or decreasing, as seen in Figure 4 in Appendix D.2. The same behavior is observed in other GAM and LR examples in Appendix D. In this example, a domain expert might conclude that the large deviation is in fact desirable because $f$ is providing varying predictions in $\mathcal{L}_{14}$ in ways that make sense, as opposed to the constant given by $f_0$. This shows that deviation maximization can work in both directions, identifying where the reference model and its assumptions may be too simplistic and giving an opportunity to improve the reference model.

**Maximum deviation vs. number of trees in a RF:** In Figure 2, we highlight one result from a set of such results in Appendix D, showing maximum deviation as a function of model complexity, here quantified by the number of estimators (trees) in a Random Forest (RF). This is a demonstration of the method in Section 4.3, which in general provides bounds on the maximum deviation. In this case, the upper ("dual") bound is informative enough to actually show a decrease as the number of estimators increases. The larger number of estimators increases averaging and may serve to make the model smoother.

## 6 Conclusion

We have considered the relationship between interpretability and safety in supervised learning through two main contributions: First, the proposal of maximum deviation as a means toward assessing safety, and second, discussion of approaches to computing maximum deviation and how these are simplified by interpretable model structure. We believe that there is much more to explore in this relationship. Appendices C and B.5 provide further discussion of several topics and future directions.

## Acknowledgements

We thank Michael Hind for several early discussions on the topic of ML model risk assessment that inspired this work, and for his overall leadership on this topic. We also thank Dhaval Patel for a discussion on the industrial assets example in Section 2.

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
