# A  Additional Problem Formulation Details

**Deviation function $D$**  For the case where the inputs $y$, $y_0$ to $D$ are real-valued scalars (which covers binary classification and regression), while Assumption 1 was stated as a sufficient condition for tractable optimization with GAMs, it is also an intuitively reasonable requirement: the deviation should increase the farther $y$ is from $y_0$ in either direction. In addition, symmetry may be desirable, i.e. $D(y, y_0) = D(y_0, y)$, to not favor one of the two models over the other. Both Assumptions 1 and 2 as well as symmetry are satisfied by monotonically increasing functions $D(|y - y_0|)$ of the absolute difference, for example powers $|y - y_0|^p$ for $p > 0$.

For the case where $y, y_0 \in \mathbb{R}^M$ as in multi-class classification, it may be advantageous for $D(y, y_0)$ to decompose into a sum over output dimensions: $D(y, y_0) = \sum_{k=1}^{M} D_k(y_k, y_{0k})$, where $y_k, y_{0k}$ are the components of $y$, $y_0$. For example, the $p$th power of the $\ell_p$ distance $\|y - y_0\|_p^p = \sum_{k=1}^{M} |y_k - y_{0k}|^p$ is separable in this manner.

**Models that abstain**  The formulation in Section 2 can also accommodate models that abstain from predicting (and possibly defer to a human expert or other fallback system). If $f(x) = \emptyset$, representing abstention, then we may set $D(\emptyset, y_0) = d$ for any $y_0 \in \mathcal{Y}$, where $d > 0$ is an intermediate value less than the maximum value that $D$ can take [49]. The value $d$ might also be less than a "typically bad" value for $D$, to reward the model for abstaining when it is uncertain.

# B  Additional Details on Deviation Maximization for Specific Model Classes

## B.1  Trees

**Rule lists**  A rule list as defined by Yang et al. [39], Angelino et al. [40] is a nested sequence of IF-THEN-ELSE statements, where the IF condition is a conjunctive rule and the THEN consequent is an output value. If each rule involves a single feature (i.e., the conjunctions are of degree 1), such rule lists are one-sided trees in the sense of Section 4.1. The number of leaves in the equivalent tree is equal to the number of rules in the list (including the last default rule).

**Intersection of $\mathcal{C}$ with a Cartesian product**  If $\mathcal{C} = \prod_{j=1}^{d} \mathcal{C}_j$ is also a Cartesian product, then determining whether the intersection is non-empty amounts to checking whether all of the coordinate-wise intersections with $\mathcal{C}_j$, $j = 1, \ldots, d$, are non-empty. If $\mathcal{C}$ is not a Cartesian product but is a union of $\ell_\infty$ balls (which are Cartesian products), then the intersection is non-empty if the intersection with any one ball is non-empty.

**Relationship between Proposition 1 and [32, Theorem 1]**  In the case of a $K = 2$-tree ensemble, [32, Theorem 1] bounds the complexity of exact robustness verification as $\min\{O(n^2), O((4n)^d\} = O(n^2)$, where $n$ is the maximum number of leaves in a tree and we assume that the feature dimension $d \geq 2$. In Proposition 1, we account for the possibly different numbers of leaves $L$ and $L_0$ in the two trees $f$ and $f_0$, and we exactly enumerate the edges, $|\mathcal{E}| \leq L_0 L \leq n^2$.

**Additive reference model**  For the case where $f$ is a decision tree and $f_0$ is a generalized additive model, if the deviation function is symmetric, $D(y, y_0) = D(y_0, y)$, then this case is covered in Section 4.2.

## B.2  Linear and Additive Models

**Categorical features**  A function $f_j(x_j)$ of a categorical feature $x_j$ can be represented in two ways, depending on whether $f$ is a GAM or a GLM. In the GAM case, we may use the native representation in which $x_j$ takes values in a finite set $\mathcal{X}_j$ of categories. In the GLM case, $x_j$ is one-hot encoded into multiple binary-valued features $x_{jk}$, one for each category $k$. Then any function $f_j$ can be represented as a linear function,

$$f_j(x_j) = \sum_{k=1}^{|\mathcal{X}_j|} w_{jk} x_{jk},$$

where $w_{jk}$ is the value of $f_j$ for category $k$.

**Implication of Assumption 1**    The second condition implies that the deviation increases or stays the same as $y$ moves away from $y_0$ in either direction.

*Proof of Proposition 2.* Let $x \in \mathcal{S}$ and $S(x) = \sum_{j=1}^{d} f_j(x_j)$. Under Assumption 1.1, if $S(x) = g(y_0)$, then

$$D(f(x), f_0(x)) = D(g^{-1}(g(y_0)), y_0) = D(y_0, y_0) = 0.$$

As $S(x)$ increases from $g(y_0)$, $f(x)$ also increases because $g^{-1}$ is an increasing function, and $D(f(x), y_0)$ increases or stays the same due to Assumption 1.2. Similarly, as $S(x)$ decreases from $g(y_0)$, $f(x)$ decreases, and $D(f(x), y_0)$ again increases or stays the same. It follows that to maximize $D(f(x), y_0)$, it suffices to separately maximize and minimize $S(x)$, compute the resulting values of $D(f(x), y_0)$, and take the larger of the two. This yields the result.    □

**Implication of Assumption 2 and identity link function** $g$    These two assumptions imply that the deviation is measured on the difference between $f$ and $f_0$ in the space in which they are additive. For example, if $f$ and $f_0$ are logistic regression models predicting the probability of belonging to one of the classes, the difference is taken in the log-odds (logit) domain. It is left to future work to determine other assumptions under which problem (1) is tractable when $f$ and $f_0$ are both additive.

**Cartesian product $\mathcal{C}$ implies Cartesian product $\mathcal{S}$**    In the cases of constant and additive $f_0$, $\mathcal{S} = \mathcal{C}$. In the decision tree case, since each leaf is a Cartesian product $\mathcal{L}_{0m} = \prod_{j=1}^{d} \mathcal{R}_{mj}$, the intersections $\mathcal{S} = \mathcal{L}_{0m} \cap \mathcal{C}$ are also Cartesian products $\prod_{j=1}^{d} \mathcal{S}_j$ where $\mathcal{S}_j = \mathcal{R}_{mj} \cap \mathcal{C}_j$.

**One-dimensional optimization complexities $C_j$**    For discrete-valued $x_j$, $C_j$ is proportional to the number of allowed values $|\mathcal{S}_j|$. For continuous $x_j$, it is common to use spline functions or tree ensembles as $f_j$ in constructing GAMs. In the former case, $C_j$ is proportional to the number of knots. In the latter, the tree ensemble can be converted to a piecewise constant function and $C_j$ is then proportional to the number of pieces. Lastly in the case where $f_j(x_j) = w_j x_j$ is linear and $\mathcal{S}_j = [\underline{X}_j, \overline{X}_j]$ is an interval, $C_j = O(1)$ because it suffices to evaluate the two endpoints.

**Convex $\mathcal{S}$**    If $\mathcal{C}$ is a convex set, then in the cases of constant and additive $f_0$, $\mathcal{S} = \mathcal{C}$ is also convex. In the case of tree-structured $f_0$, $\mathcal{S} = \mathcal{L}_{0m} \cap \mathcal{C}$ and each leaf $\mathcal{L}_{0m}$ can be represented as a convex set, with interval constraints on continuous features and set membership constraints on categorical features. The latter can be represented as $x_{jk} = 0$ constraints on the one-hot encoding (see "Categorical features" paragraph above) for non-allowed categories $k$. Hence $\mathcal{S}$ is also convex.

As a specific example, suppose that $\mathcal{S}$ is the product of independent constraints on each categorical feature and an $\ell_p$ norm constraint on the continuous features jointly. The maximization over each categorical feature has complexity $C_j = |\mathcal{S}_j|$ as noted above, while the maximization of $w^T x$ over continuous features lying in an $\ell_p$ ball has closed-form solutions for the common cases $p = 1, 2, \infty$.

**Relaxations of the Cartesian product assumption**    If the certification set $\mathcal{C}$ is not a Cartesian product, then one way to still bound the maximum deviation is to find the smallest Cartesian product $\overline{\mathcal{C}}$ that contains $\mathcal{C}$ and maximize deviation over $\overline{\mathcal{C}}$. As long as it is relatively easy to optimize linear functions over $\mathcal{C}$, then constructing such a Cartesian product is similarly easy. Another conceivable relaxation of the Cartesian product assumption is a Cartesian product of low-dimensional sets, not just one-dimensional.

### B.3    Tree Ensembles

The full algorithm for clique search from Section 4.3 is presented in Algorithm 1. It uses $Z$ as a node compatibility vector to keep track of valid leaves and $B$ a set of trees/partites not yet covered by the maximum clique. The algorithm starts with and empty clique $S$ and anytime bounds as 0. It starts the search with the smallest tree to limit the search space. This is typically $f_0$. Each leaf is added to the intermediate clique $S$ in turn (Line 6). A stronger primal bound can be achieved if the traversal is ordered in a meaningful way. In particular, starting with nodes with the highest heuristic function value $H(S)$ aids the algorithm to focus on better areas of the search space.

If the size of the clique is $K+1$ the primal bound is updated. Otherwise, the dual bound is computed. If the node is promising, the algorithm recurses to the next level. When the search is terminated at any step, the maximum deviation is bounded by $(D_{lb}, D_{ub})$.

---

**Algorithm 1** Max clique search for maximum deviation

---

**Require:** $M$ adjacency matrix, $H$ heuristic function
1: $Z[i] = 1 \forall i \in V, B = \{1, 2, \ldots, K+1\}, S = \emptyset$      $\triangleright$ All nodes valid, all trees uncovered
2: $Q = \text{Enumerate}(Z, B, S)$
3: **Initialize:** $D_{lb} = 0, D_{ub} = 0$      $\triangleright$ Anytime bounds
4: **function** Enumerate($Z, B, S$):
5:    $t = \arg\max_b \{|Z_b| \mid b \in B\}$      $\triangleright$ Uncovered tree with fewest valid nodes
6: **for** $i$ in $Z_t$ **do**
7:    $Z[i] = 0$      $\triangleright$ Mark node as incompatible
8:    $S = S \cup \{i\}$      $\triangleright$ Add to candidate clique
9:    **if** $|S| = K+1$ **then**
10:      $D_{lb} = \max(D_{lb}, D(S))$      $\triangleright$ Update primal bound
11:      $Q = Q \cup S$      $\triangleright$ Add to set of max cliques
12:      $S = S \setminus \{i\}$      $\triangleright$ Backtrack
13:    **else**
14:      $Z_{t+1} = Z_t \wedge M(i)$      $\triangleright$ Update valid nodes
15:      $B = B \setminus \{t\}$      $\triangleright$ Update uncovered trees
16:      $D_{ub} = \max(D_{ub}, H(S))$      $\triangleright$ Update dual bound
17:      **if** $D_{ub} > D_{lb}$ **then**
18:        Enumerate($Z_{t+1}, B, S$)      $\triangleright$ Recurse to next level
19:      **end if**
20:      $S = S \setminus \{i\}$      $\triangleright$ Backtrack
21:      $B = B \cup \{t\}$
22:    **end if**
23: **end for**

---

**Rule ensembles** Similar to the tree ensembles considered in Section 4.3, a rule ensemble is a linear combination of conjunctive rules, where the antecedent is a conjunction of conditions on individual features, and the consequent takes a real value if the antecedent is true and zero otherwise. They are produced by algorithms such as SLIPPER [50], that of Rückert and Kramer [51], RuleFit [41], ENDER [42] and have also been referred to as generalized linear rule models [52]. A rule ensemble can be converted into a tree ensemble by converting each conjunctive rule into an IF-THEN-ELSE rule list, which is a one-sided tree (see Appendix B.1). Specifically, the conditions in the conjunction are taken in any order, each condition is negated to become an IF condition, and the THEN consequents are all output values of zero. The final ELSE consequent, which is reached if all the IF conditions are false (and hence the original rule holds), returns the output value of the original rule. The number of leaves in the resulting tree equals the number of conditions in the conjunction plus one.

### B.4 Piecewise Lipschitz Functions

*Proof of Proposition 3.* Consider two inputs $x$ and $y$ then,

$$|h(x) - h(y)| = |(h_0 - h_1)(x) - (h_0 - h_1)(y)| = |h_0(x) - h_0(y) + h_1(y) - h_1(x)|$$
$$\leq |h_0(x) - h_0(y)| + |h_1(x) - h_1(y)| \leq c_0 \cdot \ell(x, y)^{\beta_0} + c_1 \cdot \ell(x, y)^{\beta_1}$$
$$\leq c \cdot \ell(x, y)^{\beta}$$

where, $c = 2\max(c_0, c_1)$ and $\beta = \min(\beta_0, \beta_1)$. $\qquad\square$

**Other choices for** $D(.,.)$**:** The results assumed $D(.,.)$ to be the identity function, where $\Delta = D(f_0, f)$. This choice of function clearly satisfies Assumptions 1 and 2. Again consistent with these assumptions we look at some other choices for $D(.,.)$. If $D(.,.)$ were an affine function with a positive scaling such as $D(y_0, y) = \alpha(y_0 - y) + b$ where $\alpha > 0$, then our result in equation 14 would be unchanged as only the Lipschitz constant of $\Delta$ would change, but not its (underestimated) order.

If the function were a polynomial or exponential however, no such guarantees can be made and we would be back to case 1.

### B.5 Other Model Classes: Neural Networks and Post Hoc Explanations

For model classes beyond the ones discussed in Section 4, it appears to be a greater challenge to obtain reasonably tractable algorithms that guarantee exact computation of or bounds on the maximum deviation. Here we outline some future directions for neural networks and post hoc explanations.

Robustness verification for neural networks has attracted a great deal of attention and made considerable progress, with exact approaches including satisfiability modulo theory [24] and mixed integer programming [25, 26], and incomplete methods that compute bounds using bound propagation [22, 19], linear programming and duality [18, 20, 21], and semidefinite programming [23, 27]. However, all of these methods consider a single model, effectively comparing it to a constant. Robustness verification is thus essentially a single-model case of our problem (1) in which $f_0$ is a constant (and with an appropriate choice of the deviation function $D(f, f_0)$). While we may expect that solutions to a two-model verification problem would leverage existing robustness verification methods, developing such solutions remains for future work. Furthermore, evaluation of robustness verification methods has largely been limited to local neighborhoods around input points (with typical radii $\epsilon \leq 0.1$ in terms of normalized feature values). This limitation may also need to be addressed to enable evaluation of maximum deviation in the way envisioned in this paper.

It is also natural to ask whether post hoc explanations for the model can help. One way in which this could occur is if the post hoc explanation approximates the model $f$ by a simpler model $\hat{f}$ and if the deviation function $D$ satisfies the triangle inequality $D(f(x), f_0(x)) \leq D(f(x), \hat{f}(x)) + D(\hat{f}(x), f_0(x))$. Then the maximum deviation in (1) would be bounded as

$$\max_{x \in \mathcal{C}} D(f(x), f_0(x)) \leq \max_{x \in \mathcal{C}} D(f(x), \hat{f}(x)) + \max_{x \in \mathcal{C}} D(\hat{f}(x), f_0(x)). \tag{15}$$

While we may choose $\hat{f}$ to be interpretable so that the rightmost maximization is tractable, the middle maximization asks for a *uniform* bound on the deviation between $f$ and $\hat{f}$, i.e, the fidelity of $\hat{f}$. We are not aware of a post hoc explanation method that provides such a guarantee. Indeed, in general, the middle maximization might not be any easier than the left-hand one that we set out to bound.

A (practical) possibility may be to perform quantile regression [53] for a large enough quantile to learn $\hat{f}$, as opposed to minimizing expected error as is typically done. This may be an interesting direction to explore in the future as quantile regression algorithms are available for varied model classes including linear models, tree ensembles [54] and even neural networks [55]. More investigation is needed into whether quantile regression methods can provide approximate guarantees on the middle term in (15).

Assuming that uniform proxies in the above sense can be constructed, then for certain modalities or applications it may be possible to train highly accurate proxies. For instance for tabular data, Random Forests or boosted trees might very well replicate the behavior of a neural network, in which case the machinery introduced in Section 4.3 could be used. Even for other modalities such as text and images, interpretable models such as Neural Additive Models (NAMs) [56] and continued fraction networks (CoFrNets) [57] may prove to be sufficient in some cases.

Finally, there are recent architectures such as Lipschitz neural networks [58] which are adversarially robust and hence valuable in practice. Our analysis presented in Section 4.4 for piecewise Lipschitz models would be applicable here, where the simple regret of standard bandit algorithms for a given number of queries could be reduced to (14) as opposed to (13).

## C  Further Discussion

**Worst-case approach**  The formulation of (1) as the worst case over a certification set represents a deliberate choice to depend as little as possible on a probability distribution or a dataset sampled from one. As stated in Section 2, Certification Set paragraph, $\mathcal{C}$ can depend at most on (an expanded version of) the support set of a distribution. The reason for this choice is because safety is an out-of-distribution notion: harmful outputs often arise precisely because they were not anticipated in the

data. The trade-off inherent in this choice is that the maximum deviation may be more conservative than needed. The high maximum deviation values in e.g. Figure 11 may reflect this. Given definition (1) as a starting point in this paper, future work could consider variations that depend more on a distribution and are thus less conservative, but may also offer a weaker safety guarantee.

**Choice of reference model**   The proposed definition of maximum deviation (1) depends on the choice of reference model $f_0$. Different choices will lead to different deviation values and, perhaps more importantly, different combinations of features that maximize the deviation. We have discussed possible choices in Section 2, and the results in Section 4 indicate that, as with the assessed model $f$, interpretable forms for $f_0$ can ease the computation of maximum deviation. Beyond these guidelines, it is up to ML practitioners and domain experts to decide on appropriate reference models for their application (and there may be benefit to considering more than one). We mention an additional concern with the reference model in the Ethics Discussion.

For some real applications it may be difficult to come up with a globally interpretable reference model. But specific to particular scenarios it may be possible. For instance, it might be difficult to provide general rules for how to drive a car, but in specific scenarios such as there being an obstacle in front, one can suggest that you stop or turn, which is a simple rule. So our machinery could potentially be applied at a local level where the reference model is interpretable in that locality. This might help in "spot checking" a deployed model and estimating its safety by computing these maximum deviations in scrupulously selected (challenging) scenarios.

**Impossible inputs in certification set**   Mathematically simple sets such as Cartesian products and $\ell_p$ balls permit simpler algorithms for optimizing functions over them. Accordingly, these sets have been the focus of not only the present work but also the related literature on ML verification and adversarial robustness. However, they may not serve to exclude inputs that are physically or logically impossible from the certification set $\mathcal{C}$, and thus, the resulting maximum deviation values may be too large and conservative. Here it is important to distinguish between impossible inputs and those that are merely implausible (i.e., with low probability). Techniques for capturing implausibility have been proposed for contrastive/counterfactual explanations [8, 59], whereas we expect the set of impossible inputs to be smaller and more constrained. As a simple example from the Adult Income dataset, if we agree that a wife/husband is defined to be of female/male gender (regardless of the gender of the spouse), then the cross combinations male-wife and female-husband cannot occur. Future work can consider the representation and handling of such constraints.

**White-box vs. grey-box models**   In this paper, we have assumed full "white-box" access to both $f$ and $f_0$, namely complete knowledge of their structure and parameters. Interesting questions may arise when this assumption is relaxed to different "grey-box" possibilities. For example, one could further investigate the third case in Section 4.4, where one of $f, f_0$ is black-box and the other is white-box interpretable. There may exist assumptions that we have not identified that would improve the query complexity compared to generic black-box optimization.

**Other interpretability-safety relationships**   This paper has focused on one relationship between the interpretability of a model and the safety of its outputs. It has not addressed other ways in which interpretability/explainability can affect the *risk* of a model (in the plain English sense, not the expectation of a loss function). For example, in regulated industries such as consumer finance, not providing explanations or providing inadequate ones can lead to legal, financial, and reputational risks. On the other hand, providing explanations is associated with its own risks [60]. These include the leakage of personal information or model information (intellectual property), an increase in appeals of decisions for the decision-making entity, and strategic manipulation of attributes (i.e. "gaming") by individuals to gain more favorable outcomes.

**Applicability to RL settings**   In RL, if one views the actions as labels and state representation as features, one can build a tree, albeit likely a deep/wide one, to represent exactly the RL policy, where the probability distribution over the actions can be viewed as the class distribution in a normal supervised setting. Rolling up the states, creating leaves with multiple states, and simply averaging the probabilities for each action would yield smaller trees that approximate the policy. Our work lays a foundation where in principle we can also compare $f$ and $f_0$ that are policies using such tree

representations. This may be related to a popular global explainability method [61] that samples policies and builds trees to explain them.

**Ethics**   The safety of machine learning systems has been called out by the European Commission's regulatory framework [62]. The commission states seven key dimensions to be evaluated and audited by a cross-disciplinary team: (i) human agency and oversight, (ii) technical robustness and safety, (iii) privacy and data governance, (iv) transparency, (v) diversity, non-discrimination and fairness, (vi) environmental and societal well-being, and (vii) accountability. The second of these dimensions is safety. However, Sloane et al. [63] argue that algorithmic audits are ill-defined as the underlying definitions are vague. The proposed work helps fill that ill-definedness using a quantitative approach. One may argue against this particular choice of quantification, but it does start the community down the path toward being more concrete in its definitions.

As with many other technologies, the proposed approach may be misused. For example, the reference model may be chosen in a way that hides the safety concerns of the model being evaluated. Transparent documentation and reporting with provenance guarantees can help avoid this kind of purposeful deceit [64].

# D    Experiment Details and Additional Results

## D.1    General Experiment Details

**Data processing**    We use the training set of each dataset to train models and the test set as the basis for evaluating maximum deviation, specifically as the set of centers for the $\ell_\infty$ balls in (2). Continuous features are standardized and categorical features are one-hot encoded. The $\ell_\infty$ norm is computed on the resulting normalized feature values.

**Models**    We use scikit-learn [65] to train decision tree (DT), logistic regression (LR), and Random Forest (RF) models. The corresponding complexity parameters are the number of leaves for DT (parameter `max_leaf_nodes`), the amount of $\ell_1$ regularization for LR (inverse $\ell_1$ penalty $C$), and the number of estimators/trees for RF (`n_estimators`). For additive models, we use Explainable Boosting Machines (EBM) from the InterpretML package [48] with zero interaction terms (so that the models are indeed additive). Smoothness is controlled by the `max_bins` parameter, the number of discretization bins for continuous features.

**Deviation maximization**    In all cases, when the certification set radius $r = 0$, maximum deviation can be computed simply by evaluating the models on the test set. For the case where $r > 0$, $f$ is a DT or RF, and $f_0$ is a DT, Algorithm 1 is used (on a bipartite graph if $f$ is a DT). The cases where $f$ is LR or EBM fall under the generalized additive case of Section 4.2. Given that $f_0$ is a DT, we use (7), (6) to determine the maximum deviation. For $r < \infty$ when $\mathcal{C}$ is a union of $\ell_\infty$ balls, we maximize separately over each intersection between a ball and a leaf of $f_0$ and then take the maximum over the intersections.

**Computation**    All experiments were run on CPU nodes with 64GB memory. For decision trees and tree ensembles run times of Algorithm 1 were limited to 2 hours, after which the best available bounds were used.

## D.2    Home Mortgage Disclosure Act Dataset

**Data source and pre-processing**    The data is made available by the US Consumer Finance Protection Bureau (CFPB) under the Home Mortgage Disclosure Act (HMDA). We use the national snapshot from year 2018[2] of "loan/application records," which contain information on mortgage applications and their outcomes. According to their website, the CFPB has modified the data to protect applicant and borrower privacy.

We processed the 2018 loan/application records as summarized below. These steps were informed by Gill et al. [66] but not identical to theirs:

- Restrict to complete, submitted applications with 'action_taken' $\leq 3$ (loan originated, application approved but not accepted by applicant, or application denied).
- Create a binary-valued target variable representing approval by binarizing 'action_taken' (originated or approved $\rightarrow 1$, denied $\rightarrow 0$).
- Restrict to purchases of principal residences (the most consequential in terms of people's lives, i.e., not refinances or for investment) and single-family homes.
- Restrict to loans that are not "special" in any way: conventional loans, first mortgages, not manufactured homes, no non-amortizing features, etc.
- Drop columns that are not applicable for site-built single-family homes.
- Drop columns that are not applicable or recorded until the approval/origination decision is made. For example, loan costs (points and fees) are not recorded until a loan is originated, the type of entity that purchases a loan does not apply unless the loan is originated, etc. This is in keeping with the mortgage approval scenario that we consider.
- Drop all demographic columns to reflect laws that forbid lending decisions from explicitly depending on applicant demographics.

---

[2]`https://ffiec.cfpb.gov/data-publication/snapshot-national-loan-level-dataset/` 2018

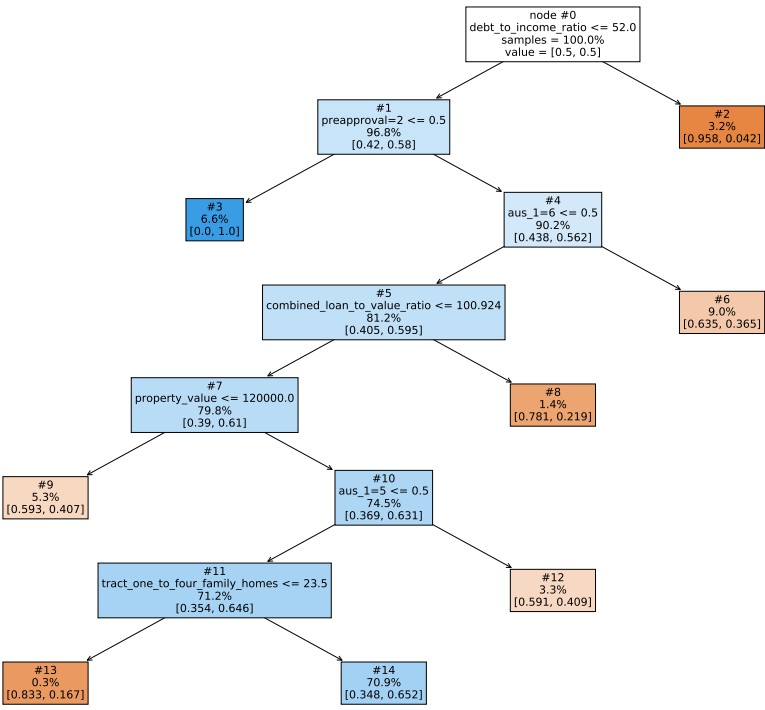

Figure 3: Decision tree reference model with 8 leaves for the HMDA dataset.

- Drop geographical columns that have too many unique values (e.g., census tract).

- Drop rows that have null or "exempt" values in key features such as loan-to-value ratio, debt-to-income ratio, property value, income.

- Take a 10% sample of the records remaining after the above processing, to make experimentation less time- and resource-consuming.

- Split the subsampled dataset 80%–20% into training and test sets.

The dataset resulting from the above processing is imbalanced, with nearly 93% of mortgage applications approved. Thus in training models, we balance the classes by weighting, either using the `class_weight='balanced'` option in scikit-learn or defining sample weights for the same purpose. On the test set, we evaluate balanced accuracy instead of accuracy.

**Reference model** Figure 3 depicts the 8-leaf DT reference model used in the experiments on the HMDA dataset. This DT has a test set balanced accuracy of $70.9\%$ and area under the receiver operating characteristic (AUC) of $0.750$. The top-level split is based on debt-to-income ratio, a measure often used in lending. Other common mortgage measures such as loan-to-value ratio, property value, and whether a preapproval was requested (value 2 means no) also appear. 'aus_1'=5 and 'aus_1'=6 refer to the automated underwriting system used to evaluate the application, with values 5 and 6 denoting "other" and "not applicable".

**LR and GAM models** In Tables 2 and 3, we show the values of inverse $\ell_1$ penalty $C$ and `max_bins` that were used for LR and GAM respectively, as well as statistics of the resulting classifiers. Test set balanced accuracy and AUC increase and reach a plateau. For LR, we take the $\ell_1$ norm of the coefficients to be the main measure of smoothness as it depends on both the number of nonzero coefficients as well as their magnitudes, which both affect the extreme values attained in (6). Since the LR balanced accuracy and AUC are not higher than those of the reference DT, we focus less on LR in what follows.

| $C$ | nonzeros | $\ell_1$ norm | bal. acc. | AUC |
|---|---|---|---|---|
| 1e-4 | 2 | 0.4 | 0.605 | 0.663 |
| 3e-4 | 7 | 1.3 | 0.633 | 0.695 |
| 1e-3 | 17 | 4.6 | 0.663 | 0.736 |
| 3e-3 | 26 | 7.7 | 0.669 | 0.744 |
| 1e-2 | 42 | 12.7 | 0.674 | 0.750 |
| 3e-2 | 68 | 16.6 | 0.675 | 0.751 |
| 1e-1 | 85 | 20.3 | 0.675 | 0.752 |
| 3e-1 | 96 | 22.2 | 0.676 | 0.752 |
| 1e+0 | 99 | 22.9 | 0.675 | 0.752 |
| 3e+0 | 100 | 23.2 | 0.675 | 0.752 |

Table 2: Number of nonzero coefficients, $\ell_1$ norm of coefficients, test set balanced accuracy, and area under the receiver operating characteristic (AUC) for logistic regression models on the HMDA dataset as a function of inverse $\ell_1$ penalty $C$.

| max_bins | bal. acc. | AUC |
|---|---|---|
| 4 | 0.669 | 0.743 |
| 8 | 0.695 | 0.772 |
| 16 | 0.711 | 0.785 |
| 32 | 0.720 | 0.798 |
| 64 | 0.723 | 0.799 |
| 128 | 0.722 | 0.800 |
| 256 | 0.722 | 0.800 |
| 512 | 0.723 | 0.800 |
| 1024 | 0.723 | 0.800 |

Table 3: Test set balanced accuracy and AUC for Explainable Boosting Machines on the HMDA dataset as a function of max_bins parameter.

For EBM, based on Table 3, we select max_bins $= 32$ as a representative model with balanced accuracy and AUC nearly equal to the maximum attainable values. Plots for this EBM model are shown in Figure 4. First note that whether a preapproval was requested (value 1 means yes) is quite predictive of final approval. The shape functions for the four continuous features debt-to-income ratio, loan-to-value ratio, property value, and income are mostly monotonic and agree with domain knowledge. The log-odds of mortgage approval decrease as debt-to-income ratio and loan-to-value ratio increase, with abrupt drops around $50\%$ for debt-to-income ratio and at $80\%$ and $100\%$ for loan-to-value ratio. For property value and income, after a minimum value is reached, the shape function increases rapidly and then stays more or less constant for high property values and incomes.

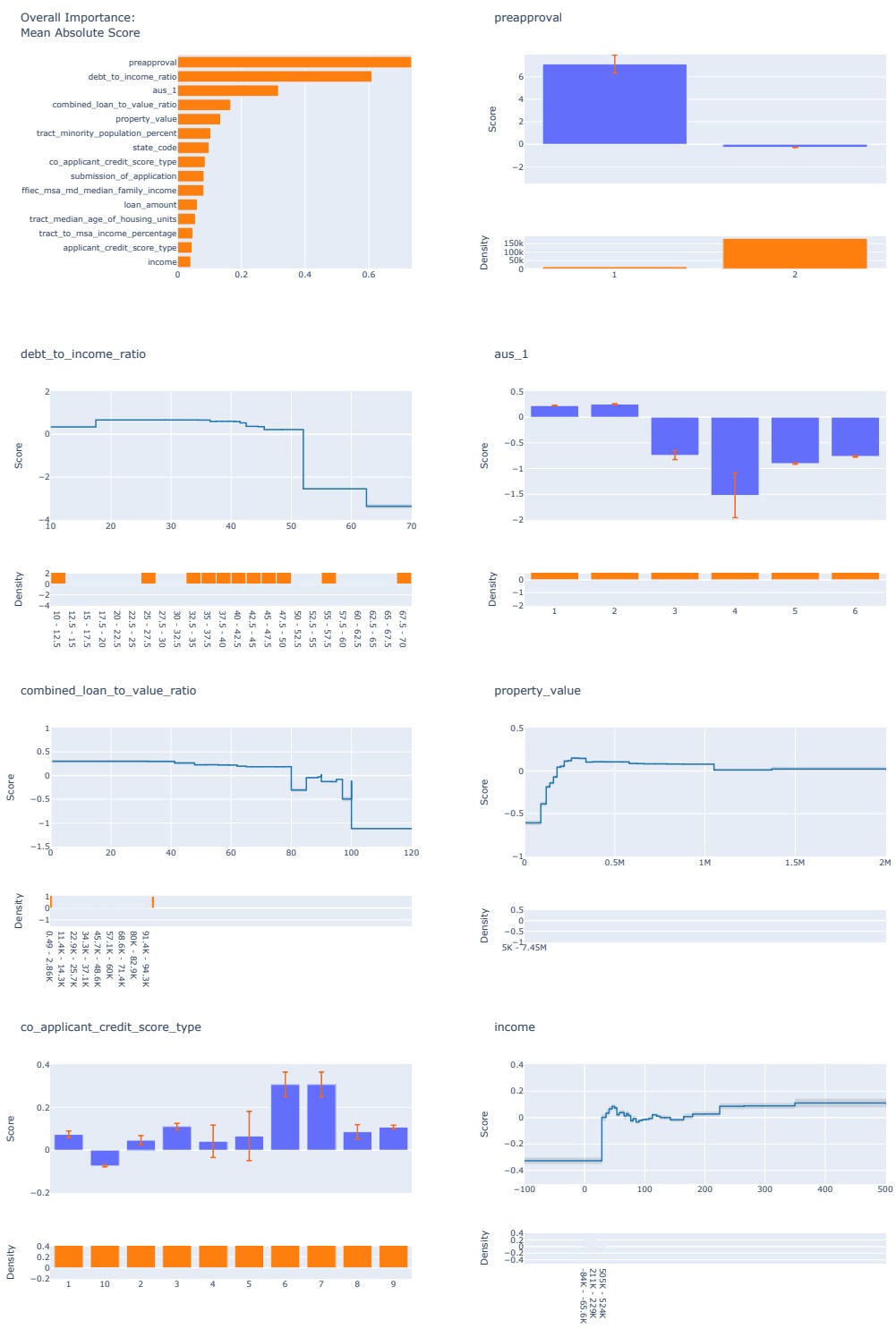

Figure 4: Feature importances and selected univariate functions $f_j$ for the Explainable Boosting Machine with `max_bins` $= 32$ on the HMDA dataset.

| $r$ | preapproval | appl_credit _score_type | loan_to_value (%) | co_appl_credit _score_type | intro_rate_period | debt_to_income (%) |
|---|---|---|---|---|---|---|
| 0.0 | 1 | 2 | 74.35 | 10 | 122.0 | 55.0 |
| 0.1 | 1 | 3 | [80. 80.] | 1 | [28.4 43.6] | [53.9 56.1] |
| 0.2 | 1 | 1 | [29.33 32.42] | 1 | [57. 60.5] | [52.8 57.2] |
| 0.4 | 1 | 7 | [ 4.34 32.42] | 7 | [329.7 360. ] | [52. 53.4] |
| 0.6 | 1 | 7 | [ 0.49 32.42] | 7 | [314.5 360. ] | [52. 55.6] |
| 0.8 | 1 | 7 | [ 0.49 32.42] | 7 | [299.3 300. ] | [52. 57.7] |
| 0.999 | 1 | 7 | [ 0.49 32.42] | 7 | [120.5 163. ] | [52. 53.9] |
| 1.001 | 1 | 6 | [ 0.49 32.42] | 6 | [120.5 159.9] | [52. 62.5] |
| 1.2 | 1 | 6 | [ 0.49 32.42] | 6 | [120.5 151. ] | [56.9 62.5] |
| 1.4 | 1 | 6 | [ 0.49 32.42] | 6 | [120.5 163. ] | [52. 57.3] |
| 1.6 | 1 | 6 | [ 0.49 32.42] | 6 | [120.5 163. ] | [52. 60.5] |
| 1.8 | 1 | 6 | [ 0.49 32.42] | 6 | [120.5 163. ] | [52. 52.7] |
| 2.0 | 1 | 6 | [ 0.49 32.42] | 6 | [120.5 163. ] | [52. 62.5] |
| $\infty$ | 1 | 6 | [ 0.49 32.42] | 6 | [120.5 163. ] | [52. 62.5] |

Table 4: Feature values that result in most positive difference in predicted probabilities between an Explainable Boosting Machine $f$ (`max_bins` $= 32$) and an 8-leaf decision tree reference model $f_0$ on the HMDA dataset. The 6 features that contribute most are shown as a function of certification set radius $r$. For radius $r > 0$, the maximizing values of continuous features form an interval because the corresponding functions $f_j$ are piecewise constant.

**Feature combinations that maximize deviation**   Table 4 shows feature values that yield the most positive difference between the predicted probabilities of the EBM $f$ with `max_bins`$= 32$ and the 8-leaf DT $f_0$. This table corresponds to the second "Conflict between $f$, $f_0$" example in Section 5. The 6 features that contribute most to the deviation are shown. These contributions are determined using (8); since the maximum deviation occurs in one of the $\ell_\infty$ ball-leaf intersections and this intersection is a Cartesian product, the feature-wise decomposition in (8) applies. The contribution of feature $j$ is then $\max_{x_j \in \mathcal{S}_j} f_j(x_j)$. We take an average of the contributions over $r$ to give a single ranking of features for all $r$. The same method is used to determine feature contributions and choose the top 6 features for Table 1.

As mentioned in Section 5, all solutions in Table 4 have 'preapproval'=1 and debt-to-income ratios $> 52\%$ that place them in leaf 2 in Figure 3. The latter results in a low predicted probability of approval from $f_0$ while the former makes a large positive contribution to the probability from $f$ (see Figure 4). The values of the other features also make increasingly larger positive contributions to $f$ as $r$ increases. Loan-to-value ratio decreases, while co-applicant credit score type moves from type 10 (no co-applicant, hence weaker application) to increasingly favorable score types (1, 7, 6, see Figure 4); applicant credit score is similar. This behavior is similar to the movement toward extreme points seen in Table 1.

**Dependence on model complexity**   Figure 5 shows the dependence of maximum deviation on the complexity of model $f$, quantified by the number of leaves for DTs, coefficient $\ell_1$ norm for LR, `max_bins` for EBM, and the number of estimators (trees) for RF. The DT and RF cases demonstrate the methods in Section 4.3, specifically Algorithm 1, where in the RF case, the algorithm may only provide bounds after a time limit of two hours. The plots show that maximum deviation may or may not increase with model complexity. In Figure 5a, the deviation is small for a 10-leaf DT and increases rapidly. Figures 5b and 5c indicate that maximum deviation is sensitive to the $\ell_1$ norm of LR models but not to the `max_bins` parameter of EBMs. The latter may increase the resolution of the EBM shape functions but not their dynamic range.

**Dependence on certification set size**   Figure 6 shows the dependence of maximum deviation on the certification set radius $r$. For LR and GAM, the maximum deviation is greater for $r > 0$ than for $r = 0$, showing that evaluation on a finite test set may not be sufficient and infinite certification sets (with $r > 0$) should be considered, especially to account for unexpected, out-of-distribution deviations. There are jumps at $r = 1$ because this is the radius that permits values of categorical features of test set points (the ball centers in (2)) to change to any other value. In the case of GAM,

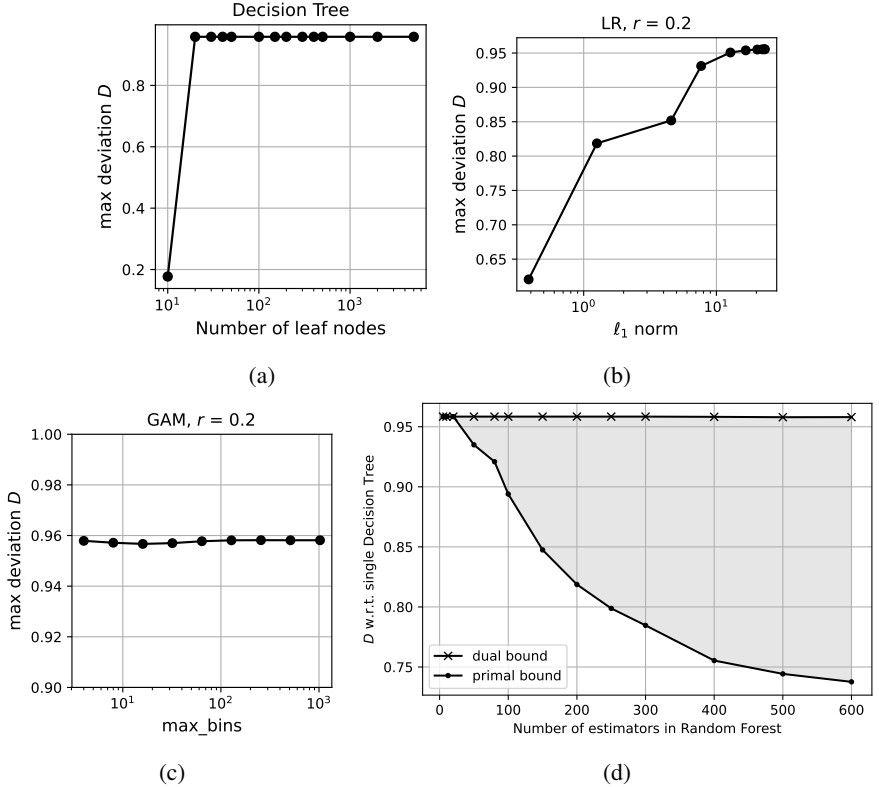

Figure 5: Maximum deviation $D$ on the HMDA dataset as a function of model complexity for (a) DT (number of leaves), (b) LR ($\ell_1$ norm), (c) GAM (`max_bins`), and (d) RF (number of estimators).

this jump is sufficient for the deviation to equal that for $r = \infty$ ($\mathcal{C} = \mathcal{X}$, dashed line in figure). On the other hand, the deviation for DT and RF remains constant as a function of $r$.

**Running time** Figure 7 shows the time required to compute the maximum deviation for LR and GAM on the HMDA dataset. These times were obtained using a single 2.0 GHz core of a server with 64 GB of memory (only a small fraction of which was used) running Ubuntu 16.04 (64-bit). The times increase with the $\ell_\infty$ ball radius $r$ because of the increasing number of ball-leaf intersections that become non-empty and hence need to be evaluated. The time for $r = 0$ is minimal because this case requires only model evaluation over the finite test set, as mentioned. The jumps at $r = 1$ are due again to the ability of categorical features to change values, leading to an increase in ball-leaf intersections. The filled-in regions show that there was little variation due to different $\ell_1$ norms for LR or `max_bins` for GAM. This was most likely because of a vectorized implementation, which operates on all LR coefficients or all GAM bins at once (i.e., without a for loop).

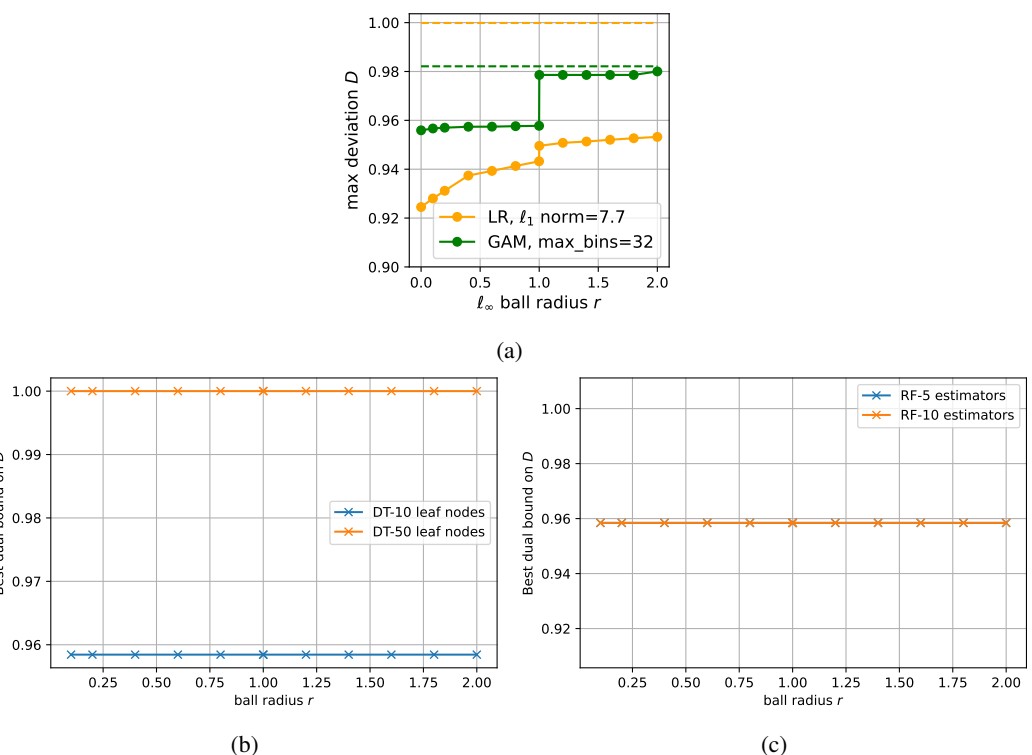

Figure 6: Maximum deviation $D$ on the HMDA dataset as a function of certification set radius $r$ for (a) LR and GAM, (b) DT (10 and 50 leaves), (c) RF (5 and 10 estimators). Dashed lines in (a) indicate the $r \to \infty$ asymptote of the curve of the same color.

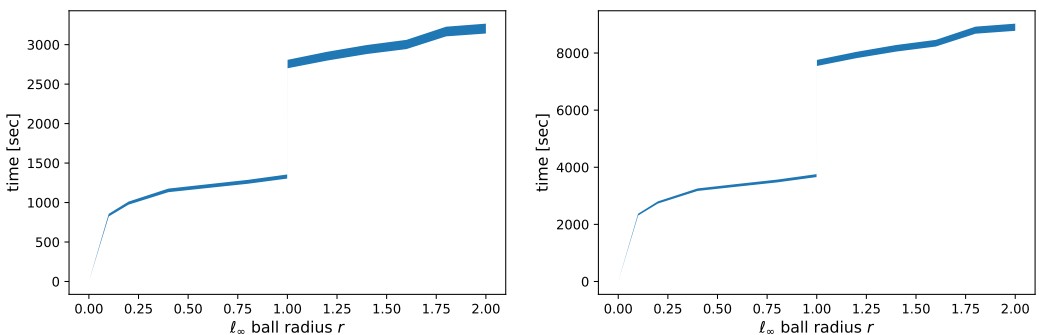

Figure 7: Time to compute maximum deviation for logistic regression models (left) and Explainable Boosting Machines (right) on the HMDA dataset as a function of certification set size (radius $r$). The filled-in region shows the min-max variation with model complexity ($\ell_1$ norm for LR, `max_bins` for EBM).

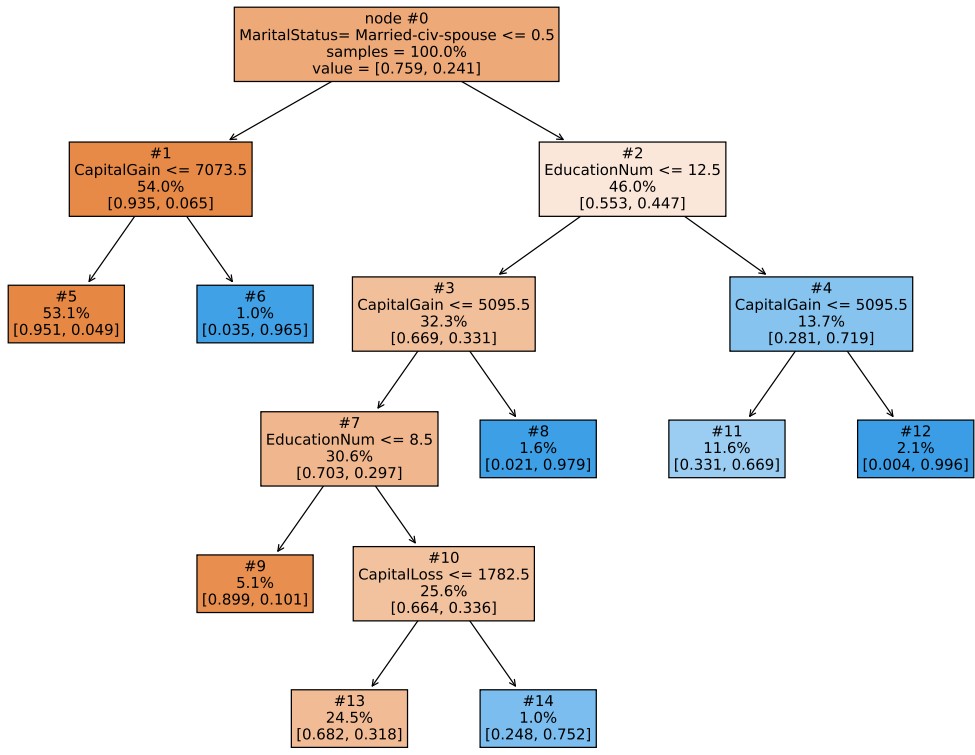

Figure 8: Decision tree reference model with 8 leaves for the Adult Income dataset.

### D.3 Adult Income dataset

We use the given partition of the Adult Income dataset into training and test sets.

**Reference model**  Figure 8 depicts the 8-leaf DT reference model used in the experiments on the Adult Income dataset. This DT has 85.0% accuracy on the test set. The root node separates individuals based on whether the marital status is *Married-civ-spouse*. The remaining splits divide the population into those with high and low education, high and low capital gains, and high and low capital losses. In particular, having high capital gains or losses is a good predictor of high income ($> \$50000$).

**LR and GAM models**  Tables 5 and 6 show the values of $C$ and `max_bins` used for LR and GAM respectively together with statistics of the resulting classifiers. Based in part on Tables 5 and 6, we select $C = 0.01$ and `max_bins` $= 8$ as representative models that remain simple and have accuracies and AUCs not far from the maximum attainable. Plots for these two models are shown in Figures 9 and 10.

| $C$ | nonzeros | $\ell_1$ norm | accuracy | AUC |
|---|---|---|---|---|
| 3e-4 | 1 | 0.2 | 0.764 | 0.715 |
| 1e-3 | 6 | 2.6 | 0.825 | 0.885 |
| 3e-3 | 7 | 4.7 | 0.840 | 0.895 |
| 1e-2 | 16 | 7.4 | 0.848 | 0.900 |
| 3e-2 | 30 | 12.9 | 0.852 | 0.905 |
| 1e-1 | 38 | 17.3 | 0.853 | 0.905 |
| 3e-1 | 62 | 25.3 | 0.853 | 0.905 |
| 1e+0 | 83 | 40.7 | 0.852 | 0.905 |
| 3e+0 | 92 | 54.6 | 0.852 | 0.905 |
| 1e+1 | 101 | 65.1 | 0.852 | 0.904 |
| 3e+1 | 105 | 73.5 | 0.852 | 0.904 |
| 1e+2 | 107 | 77.6 | 0.852 | 0.904 |
| 3e+2 | 107 | 79.1 | 0.852 | 0.904 |

Table 5: Number of nonzero coefficients, $\ell_1$ norm of coefficients, test set accuracy, and AUC for logistic regression models on the Adult Income dataset as a function of inverse $\ell_1$ penalty $C$.

| max_bins | accuracy | AUC |
|---|---|---|
| 4 | 0.858 | 0.910 |
| 8 | 0.862 | 0.915 |
| 16 | 0.865 | 0.920 |
| 32 | 0.870 | 0.924 |
| 64 | 0.871 | 0.925 |
| 128 | 0.871 | 0.925 |
| 256 | 0.872 | 0.925 |
| 512 | 0.871 | 0.925 |
| 1024 | 0.871 | 0.925 |

Table 6: Test set accuracy and AUC for Explainable Boosting Machines on the Adult Income dataset as a function of max_bins parameter.

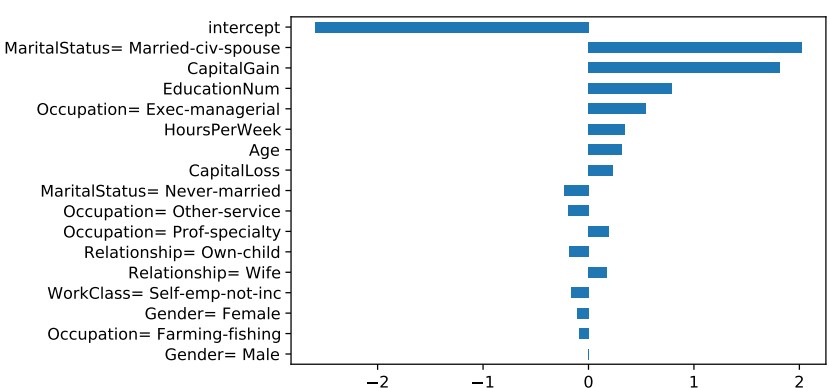

Figure 9: Coefficient values of the logistic regression model with $C = 0.01$ (16 nonzeros) for the Adult Income dataset.

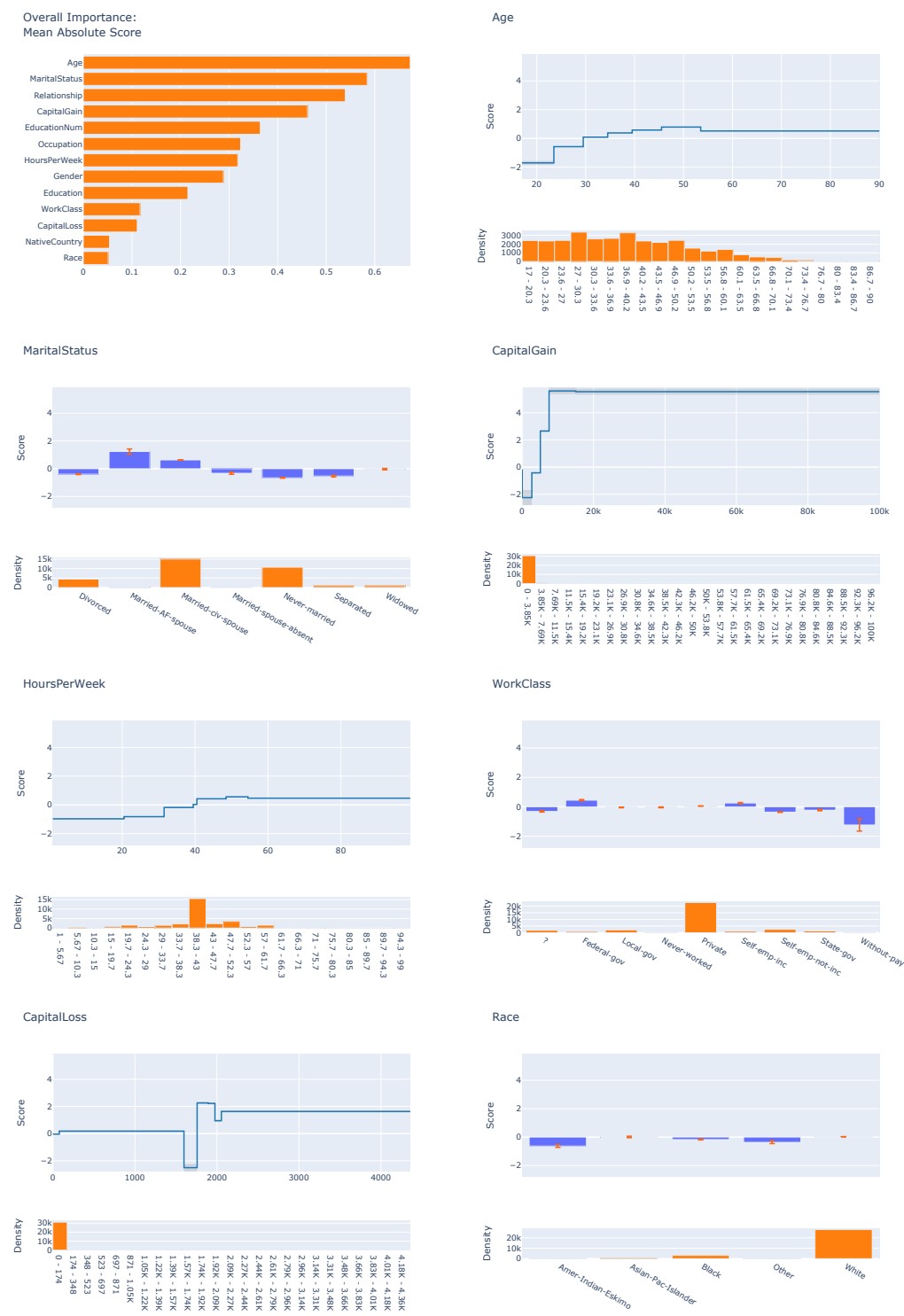

Figure 10: Feature importances and selected univariate functions $f_j$ for the Explainable Boosting Machine with `max_bins` = 8 on the Adult Income dataset.

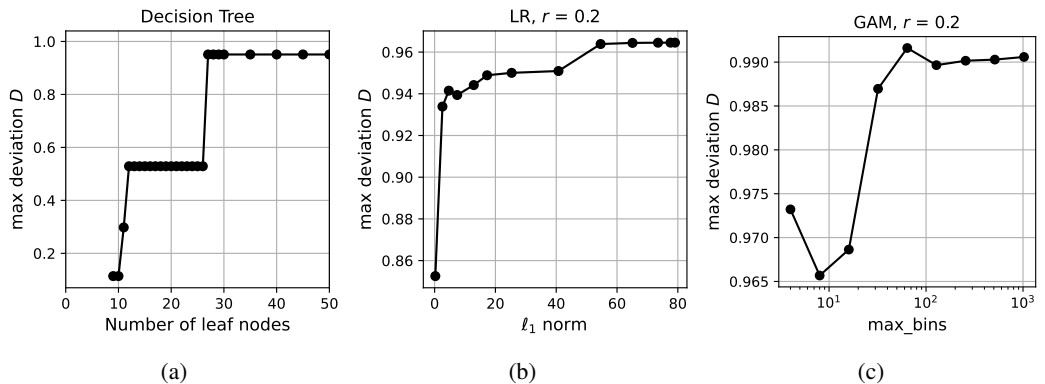

(a)                                  (b)                                  (c)

Figure 11: Maximum deviation $D$ on the Adult Income dataset as a function of model complexity for (a) DT (number of leaves), (b) LR ($\ell_1$ norm), and (c) GAM (`max_bins`).

**Dependence on model complexity**    Figure 11 shows the dependence on model complexity for DT (number of leaves), LR (coefficient $\ell_1$ norm), and EBM (`max_bins`). In Figure 11a, the maximum deviation is $0.114$ for trees with $9$ and $10$ leaves, and remains moderate up to $26$ leaves, which is different than in Figure 5a. Similar to Figures 5b and 5c, $\ell_1$ norm has a larger effect on maximum deviation than `max_bins` (note the vertical scale in Figure 11c).

**Dependence on certification set size**    Figure 12a shows the dependence on the certification set radius $r$ for DT, LR, and GAM. The patterns are similar to those in Figure 6: the deviations for LR and GAM increase from $r = 0$ and have jumps at $r = 1$, while the deviation for DT remains constant. One difference is that the LR curve in Figure 12a meets its $r \to \infty$ asymptote (dashed line in figure), similar to GAM.

Figure 12b shows the upper bound on the maximum deviation as a function of the certification set size for two RF models. As the test set is large in this case, the deviations observed even for small values of $r$ are high and grow to reach the value of the full feature space quickly.

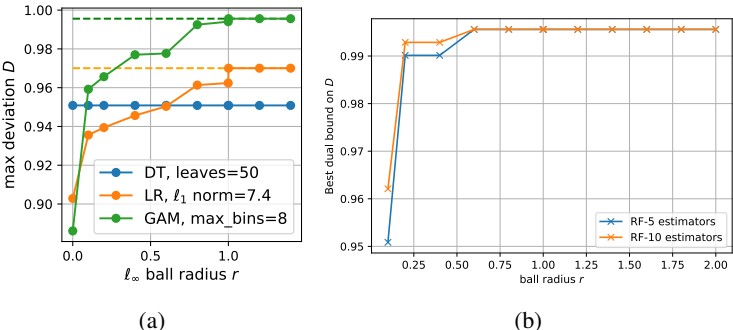

(a)                                                      (b)

Figure 12: (Left) Maximum deviation $D$ on the Adult Income dataset as a function of certification set radius $r$ for DT, LR, and GAM. (Right) Upper bound on maximum deviation of $f$, a Random Forest, trained on the Adult Income dataset.

**Relationships with accuracy and robust accuracy**    In Figure 13, we show maximum deviation as a function of test set accuracy for the DT, LR, and GAM models shown in Figure 11 (the LR and GAM models are listed in Tables 5 and 6). Broadly, the plots show two regimes: one where accuracy increases and maximum deviation increases moderately or not at all, and one where accuracy stalls while maximum deviation increases. The latter is less desirable as it suggests increasing safety risks without a gain in accuracy. The last branch of the DT curve actually decreases in accuracy, indicating overfitting, while maximum deviation is high.

We also consider the relationship of maximum deviation to *robust accuracy*. Following Wong and Kolter [20], robust loss for a pair $(x, y)$ is defined as the worst-case loss over an $\ell_\infty$ ball centered at

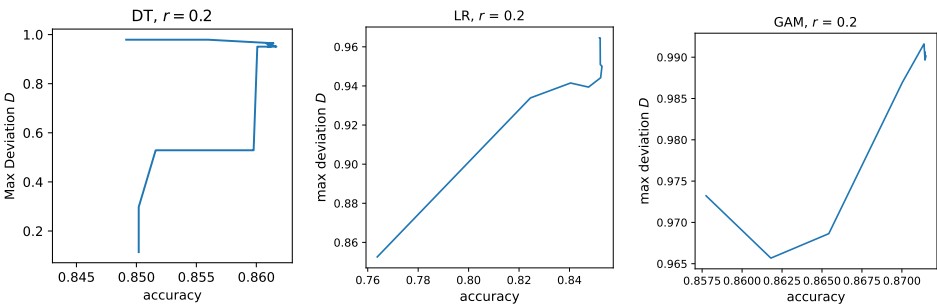

Figure 13: Maximum deviation $D$ (at certification set radius $r = 0.2$) vs. test set accuracy on the Adult Income dataset.

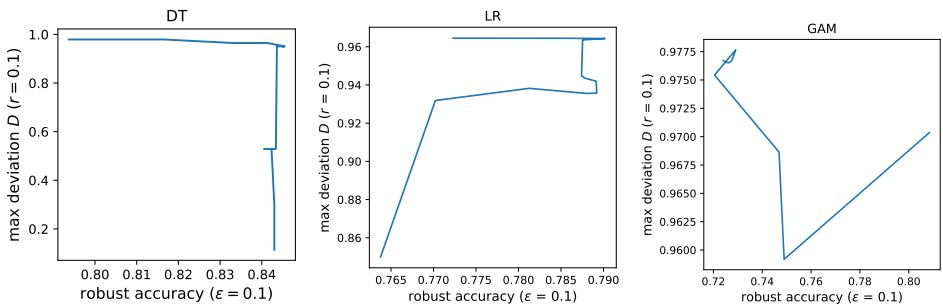

Figure 14: Maximum deviation $D$ vs. robust accuracy ($\epsilon = 0.1$, $r = 0.1$) on the Adult Income dataset.

$x$,

$$\max_{\|\Delta\|_\infty \leq \epsilon} L(f(x + \Delta), y), \tag{16}$$

and robust accuracy is therefore 1 minus the average robust 0-1 loss over a dataset. While Wong and Kolter [20] focus on bounding robust loss for feedforward neural networks with ReLU activations, we find that the results in Section 4.2 apply to computing robust loss (16) exactly for LR and GAM models. Specifically, for 0-1 loss and $\ell_\infty$ balls, the separable optimization (8) applies, and the worst case is obtained by minimizing $f$ when the label $y$ is positive and maximizing $f$ when $y$ is negative.

The resulting robust accuracy values for DT, LR and GAM are plotted in Figure 14 in a similar fashion as Figure 13. Here we set $\epsilon = 0.1$ and $r = 0.1$ as well in computing maximum deviation. The DT plot shows maximum deviation increasing with model complexity while robust accuracy is stable up to a point. In the subsequent regime, when there are a large number of leaves, model robustness reduces while deviation remains high. The LR plot begins similarly to the one in Figure 13 in that robust accuracy increases along with maximum deviation, but then it stalls and decreases for maximum deviation above $0.94$. In the GAM plot, robust accuracy actually decreases with the `max_bins` parameter, i.e., the curve goes from right to left.

**Breakdown by leaves of** $f_0$ In Figures 15–18, we plot the maximum log-odds achieved by model $f$ (max on RHS of (7)), the minimum log-odds achieved by $f$, and the reference model log-odds $g(y_{0m})$ over each leaf of the decision tree reference model in Figure 8. Plots are on the log-odds scale to show the deviations more clearly, including those that would be compressed by the nonlinear logistic function $g^{-1}(z) = 1/(1 + e^{-z})$. Figures 15 and 17 show the dependence on the certification set radius $r$ while Figures 16 and 18 show dependence on the smoothness parameters for LR and GAM. These figures provide a more granular picture corresponding to the summary in Figure 11 and support the trends seen there. In Figures 15 and 17, there are jumps at $r = 1$ because this is the smallest radius that permits the values of categorical features of test set points (the ball centers in (2)) to change to any other value. (Recall that categorical features are one-hot encoded into binary-valued features.) In Figure 17, the GAM achieves the limiting deviations corresponding to $r = \infty$ ($\mathcal{C} = \mathcal{X}$,

dashed lines) no later than $r = 1.2$. In Figure 15, the LR model achieves the lower limit on log-odds as soon as $r > 1$ but the upper limit is not achieved for most leaves.

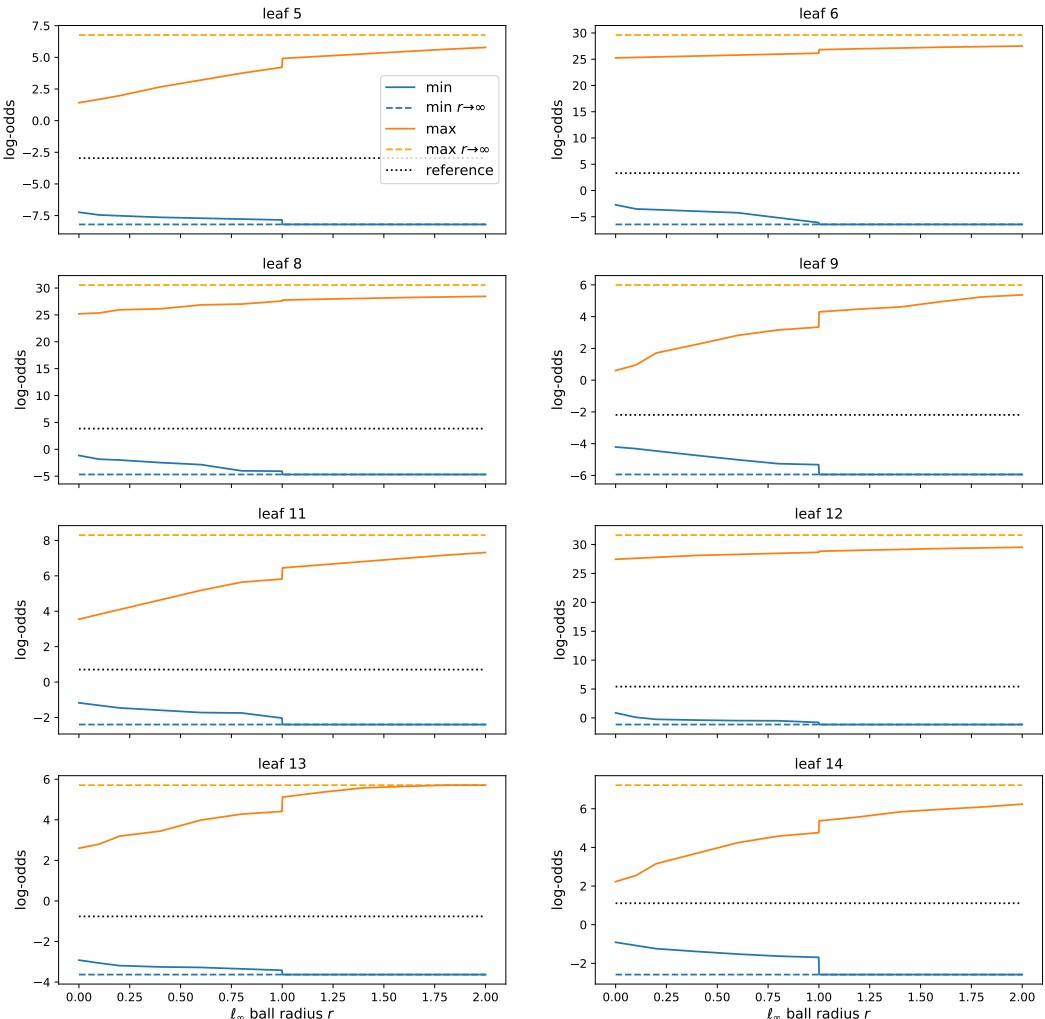

Figure 15: Minimum and maximum predicted log-odds for a logistic regression model with inverse $\ell_1$ penalty $C = 0.01$, as a function of certification set size (radius $r$) and broken down by leaves of the decision tree reference model.

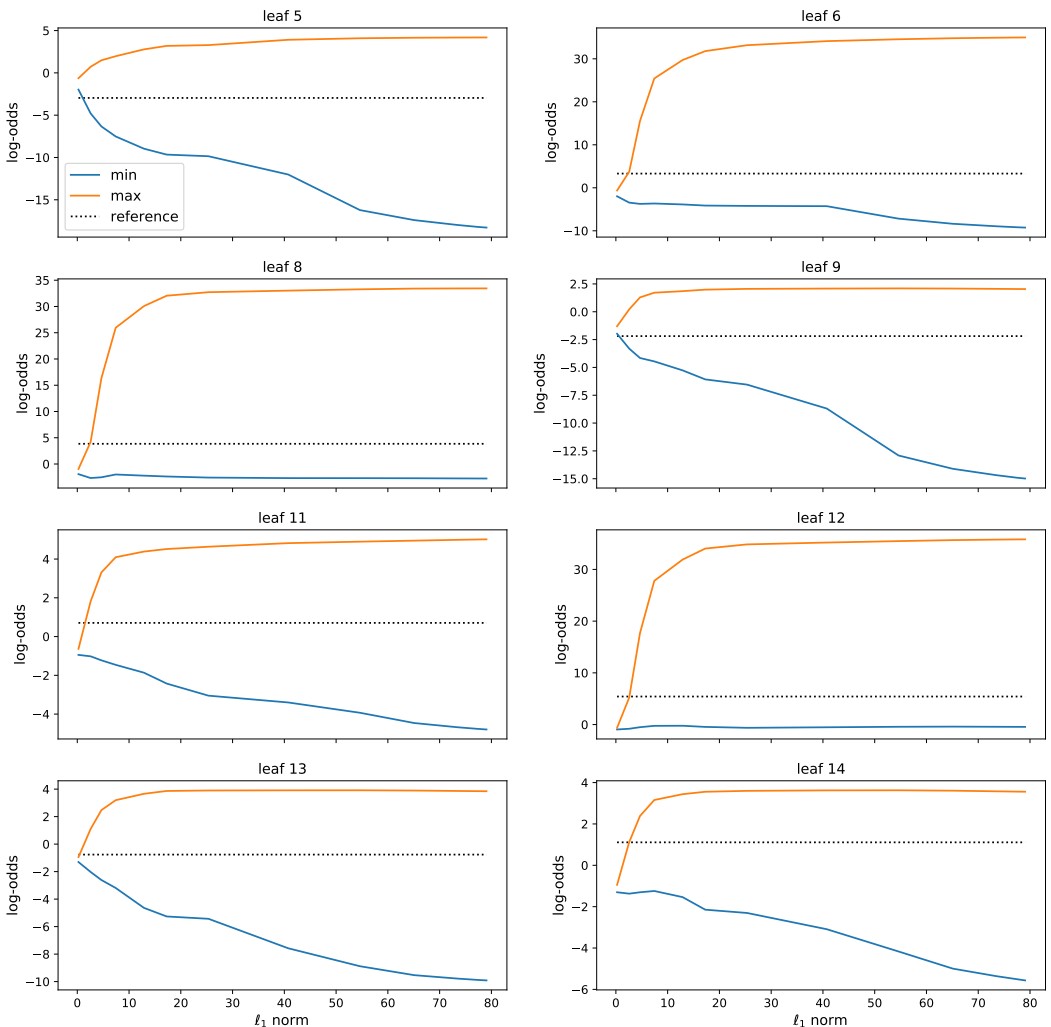

Figure 16: Minimum and maximum predicted log-odds for logistic regression models with different $\ell_1$ penalties $C$, broken down by leaves of the decision tree reference model. The certification set $\ell_\infty$ ball radius is $r = 0.2$.

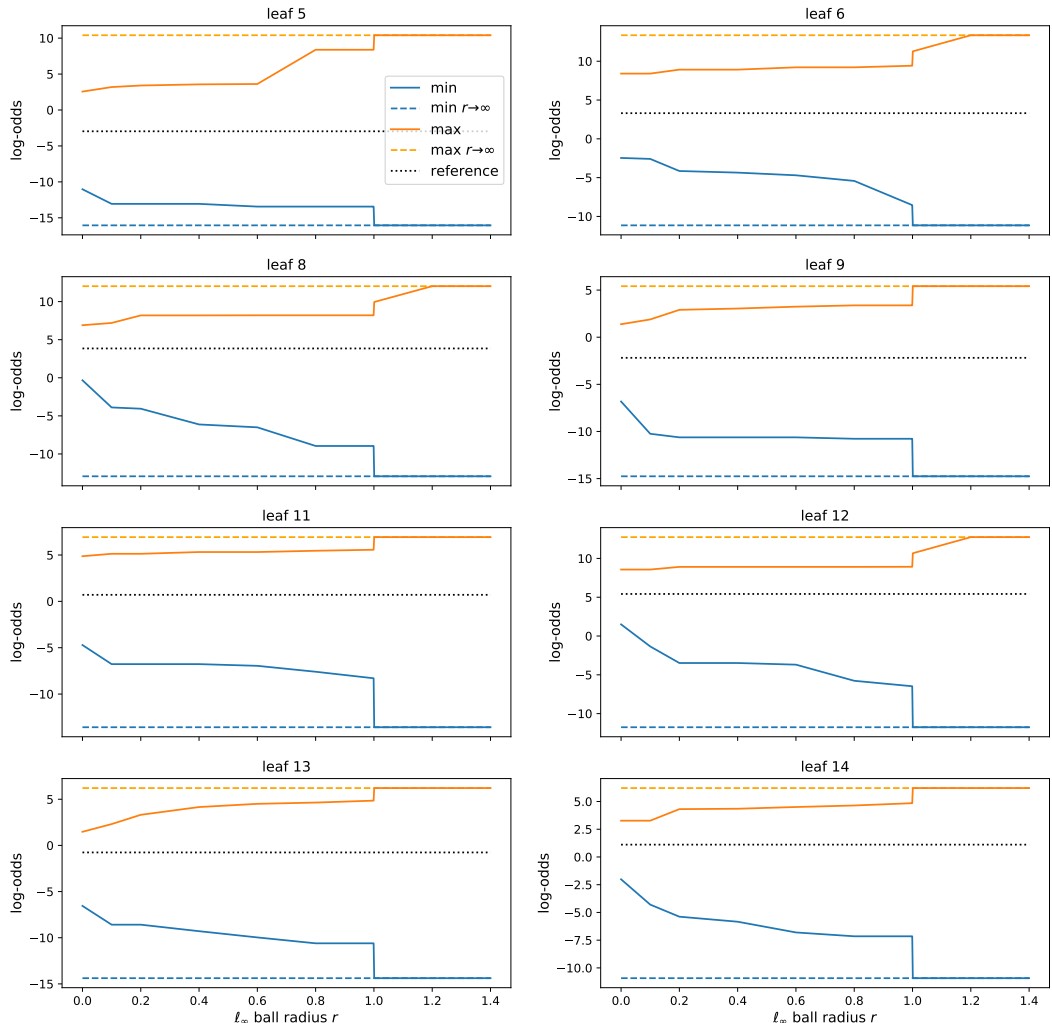

Figure 17: Minimum and maximum predicted log-odds for an Explainable Boosting Machine with max_bins $= 8$, as a function of certification set size (radius $r$) and broken down by leaves of the decision tree reference model.

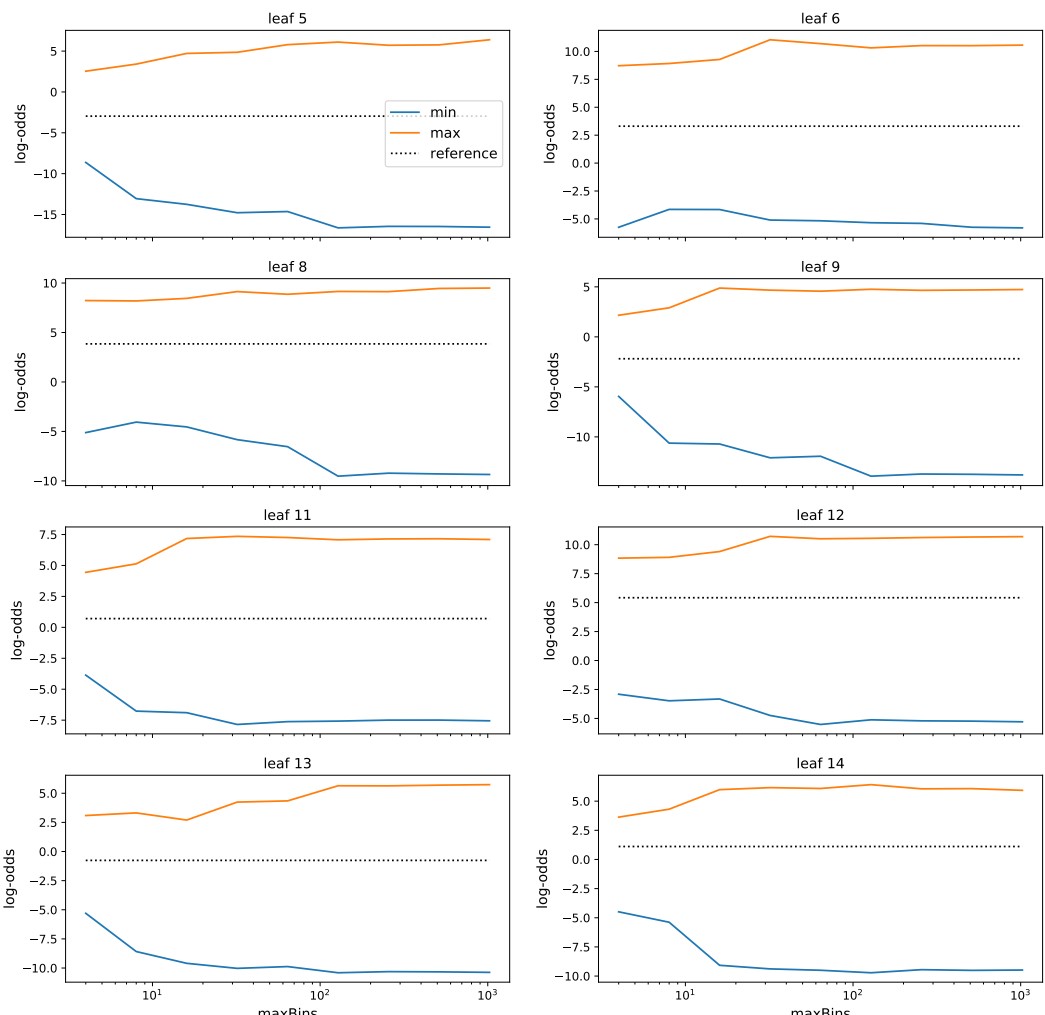

Figure 18: Minimum and maximum predicted log-odds for Explainable Boosting Machines with different max_bins values, broken down by leaves of the decision tree reference model. The certification set $\ell_\infty$ ball radius is $r = 0.2$.

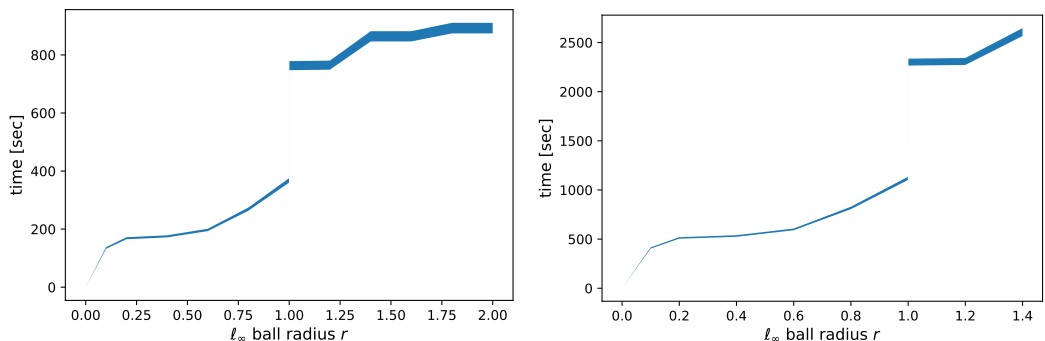

Figure 19: Time to compute maximum deviation for logistic regression models (left) and Explainable Boosting Machines (right) on the Adult Income dataset as a function of certification set size (radius $r$). The filled-in region shows the min-max variation with model complexity ($\ell_1$ norm for LR, `max_bins` for EBM).

**Running time**   Figure 19 shows the time required to compute the maximum deviation for LR and GAM on the Adult Income dataset. The same observations apply as in Figure 7 earlier.

**Primal bounds for RF**   The primal bound for max-deviation shown in Figure 2 for RF is updated each time the algorithm finds a $K + 1$ partite, i.e. has examined all trees in the Random Forest. The max-deviation computed for such a partite is a valid deviation. To prove if its optimal, Algorithm 1 needs to run to completion which may not be feasible. Figure 2 shows that as the RF models get larger (number of partites increase), it gets harder to find primal solutions.

**Maximal cliques evaluated for DT, RF**   To investigate the effectiveness of pruning by bounds in Algorithm 1, we investigate the number of times all the decision trees in the Random Forest have to be processed. This represents number of times the state could not be pruned and needed to be evaluated fully.

Figure 20 shows two aspects at play. (a) Pruning by bound is effective in restricting the search space more so for Random Forests than for decision trees, and (b) for larger graphs, more time is spent in computing bounds in Eq. (11).

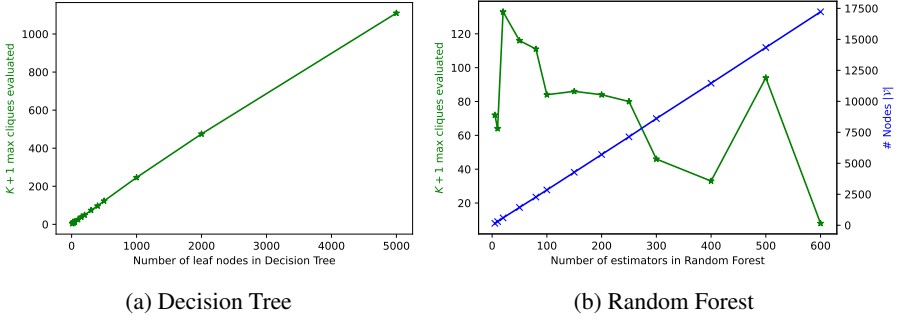

(a) Decision Tree                  (b) Random Forest

Figure 20: Effectiveness of pruning by bound for tree-based models

**Feature combinations that maximize deviation**   In Tables 7 and 8, we report feature values that maximize deviation over selected leaves of the DT reference model, for LR and GAM respectively. For Table 7, we have chosen the minimum log-odds over leaf 6 (corresponding to Figure 15, leaf 6, blue curve), which is one of the two leaves $m$ in (7) that maximize the deviation overall (the other being leaf 8). For Table 8, the minimum log-odds over leaf 12 is chosen (corresponding to Figure 17, leaf 12, blue curve) because this maximizes the deviation overall for most values of $r$. The tables show the 6 features that contribute most to the minimum log-odds. These contributions are again determined using (8) (with max replaced by min); since the minimum log-odds occurs in one of

the $\ell_\infty$ ball-leaf intersections and this intersection is a Cartesian product, the decomposition in (8) applies. The contribution of feature $j$ is then $\min_{x_j \in S_j} f_j(x_j)$. As in Tables 1 and 4, we take an average of the contributions over $r$ to give a single ranking of features for all $r$.

| $r$ | EducationNum | HoursPerWeek | Age | MaritalStatus | Occupation | Relationship |
|---|---|---|---|---|---|---|
| 0.0 | 9.0 | 15.0 | 23.0 | Never-married | Sales | Own-child |
| 0.1 | 4.7 | 33.8 | 26.6 | Never-married | Transport-moving | Not-in-family |
| 0.2 | 4.5 | 32.5 | 25.3 | Never-married | Transport-moving | Not-in-family |
| 0.4 | 4.0 | 30.1 | 22.5 | Never-married | Transport-moving | Not-in-family |
| 0.6 | 3.5 | 27.6 | 19.8 | Never-married | Transport-moving | Not-in-family |
| 0.8 | 1.0 | 26.1 | 17.0 | Never-married | Farming-fishing | Not-in-family |
| 0.999 | 1.0 | 7.7 | 17.0 | Never-married | Other-service | Own-child |
| 1.001 | 1.0 | 1.0 | 17.0 | Never-married | Other-service | Own-child |
| 1.2 | 1.0 | 1.0 | 17.0 | Never-married | Other-service | Own-child |
| 1.4 | 1.0 | 1.0 | 17.0 | Never-married | Other-service | Own-child |
| 1.6 | 1.0 | 1.0 | 17.0 | Never-married | Other-service | Own-child |
| 1.8 | 1.0 | 1.0 | 17.0 | Never-married | Other-service | Own-child |
| 2.0 | 1.0 | 1.0 | 17.0 | Never-married | Other-service | Own-child |
| $\infty$ | 1.0 | 1.0 | 17.0 | Never-married | Other-service | Own-child |

Table 7: Feature values that minimize log-odds for a logistic regression model ($C = 0.01$) over leaf 6 of the decision tree reference model. The 6 features that contribute most to the minimum are shown as a function of certification set radius $r$.

As $r$ increases, the predominant trend of the values of continuous features is toward extremes of the domain $\mathcal{X}$, depending on the sign of the corresponding LR coefficient $w_j$ or shape of the GAM function $f_j$. For example, EducationNum (education on an ordinal scale), hours per week, and age decrease toward minimum values, while capital gain occupies the minimal interval permitted for leaf 12 (see Figure 8). (These examples make sense since the log-odds of high income is being minimized.) This movement toward extremes is expected in the LR case because the functions $w_j x_j$ are either increasing or decreasing, and it is also true for GAM if the function $f_j$ is mainly increasing or decreasing. The values sometimes change abruptly in the opposite direction, for example hours per week in both Tables 7, 8, and age in the latter. These abrupt changes are due to the minimum jumping from one ball in (2) to another as $r$ increases, but the overall trend eventually prevails. For categorical features, the trend is toward values that minimize $f_j(x_j)$, e.g., *Never-married* marital status, *Without-pay* work class. While the contribution of each of these features to minimizing log-odds may be limited, together they do add up.

| $r$ | CapitalLoss | Age | HoursPerWeek | WorkClass | CapitalGain | Race |
|---|---|---|---|---|---|---|
| 0.0 | 0 | 29.0 | 40.0 | Private | 7298 | White |
| 0.1 | [ 0 40] | [53.6 56.4] | [18.8 20.5] | ? | [5095 5119] | White |
| 0.2 | [ 0 78] | [23.3 23.5] | [4.5 9.5] | State-gov | [5095 5119] | White |
| 0.4 | [ 0 78] | [20.5 23.5] | [ 2.1 11.9] | State-gov | [5095 5119] | White |
| 0.6 | [ 0 78] | [28.8 29.5] | [30.6 31.5] | Private | [5095 5119] | Asian-Pac-Islander |
| 0.8 | [1598 1759] | [20.1 23.5] | [30.1 31.5] | State-gov | [5095 5119] | White |
| 0.999 | [1598 1759] | [21.4 23.5] | [27.7 31.5] | Private | [5095 5119] | Amer-Indian-Eskimo |
| 1.001 | [1598 1759] | [17. 23.5] | [11.6 20.5] | Without-pay | [5095 5119] | Amer-Indian-Eskimo |
| 1.2 | [1598 1759] | [17. 23.5] | [ 9.2 20.5] | Without-pay | [5095 5119] | Amer-Indian-Eskimo |
| 1.4 | [1598 1759] | [17. 23.5] | [ 6.7 20.5] | Without-pay | [5095 5119] | Amer-Indian-Eskimo |
| $\infty$ | [1598 1759] | [17. 23.5] | [ 1. 20.5] | Without-pay | [5095 5119] | Amer-Indian-Eskimo |

Table 8: Feature values that minimize log-odds for an Explainable Boosting Machine (`max_bins` $= 8$) over leaf 12 of the decision tree reference model. The 6 features that contribute most to the minimum are shown as a function of certification set radius $r$. For $r > 0$, the minimizing values of continuous features form an interval because the corresponding functions $f_j$ are piecewise constant.

A notable exception to the trend toward extremes is capital loss in Table 8. This was discussed in the "Identification of an artifact" example in Section 5.

Given the results in Tables 7 and 8, one question that arises is whether the feature combinations are indeed possible, if not the ones for $r \to \infty$, then at least for some finite value of $r$. For the top

features shown in the two tables, while some combinations may appear improbable (for example, EducationNum = 1 and 1 hour per week), we submit that none appear *impossible*. However, if one considers features beyond the top 6, then some "impossible" combinations do occur (e.g., a female husband), although the contributions of these features to the minimum log-odds are much less. We touch upon this issue in Appendix C.

The next question one might consider is the implication of these maximal deviations. From Figure 8, it is seen that leaf 6 classifies individuals with high capital gains as high income with high probability (0.965). Leaf 12 adds the attributes of married status and high education, and hence classifies as high income with even higher probability (0.996). At the same time, the feature values in Tables 7 and 8, which minimize log-odds for LR and GAM, also make sense according to basic domain knowledge. For example, few hours per week and young age are associated with lower income, as are *Without-pay* work class and *Amer-Indian-Eskimo* race in the United States. When these conflicting associations occur in combination and the combination does not appear impossible, the question may be which one prevails. Such a question might be resolvable by a domain expert. Alternatively, the disagreement between models $f$ and $f_0$ on the extreme examples in Tables 7, 8 may be reason to be cautious about using either of the models in these cases. This might lead to a way of combining the models or abstaining from prediction altogether. Lastly, the anomalously low region in the CapitalLoss function identified in Table 8 is a clear, concrete example where further investigation is warranted.

## D.4 Lending Club dataset

This dataset consists of 2.26 million rows with 14 features on loans. The target variable is whether a loan will be paid-off or defaulted on. Features describe the terms of the loan, e.g. duration, grade, purpose, etc. and borrower financial information such as credit history and income. For this case study, we consider a loan approval scenario using only information available at the time of application. In particular, we exclude the feature 'total_pymnt' (total payment over time on the loan), which becomes known at essentially the same time as the target variable. (When 'total_pymnt' is included as a feature, the prediction task becomes easy and accuracies in the high $90\%$ range are possible.)

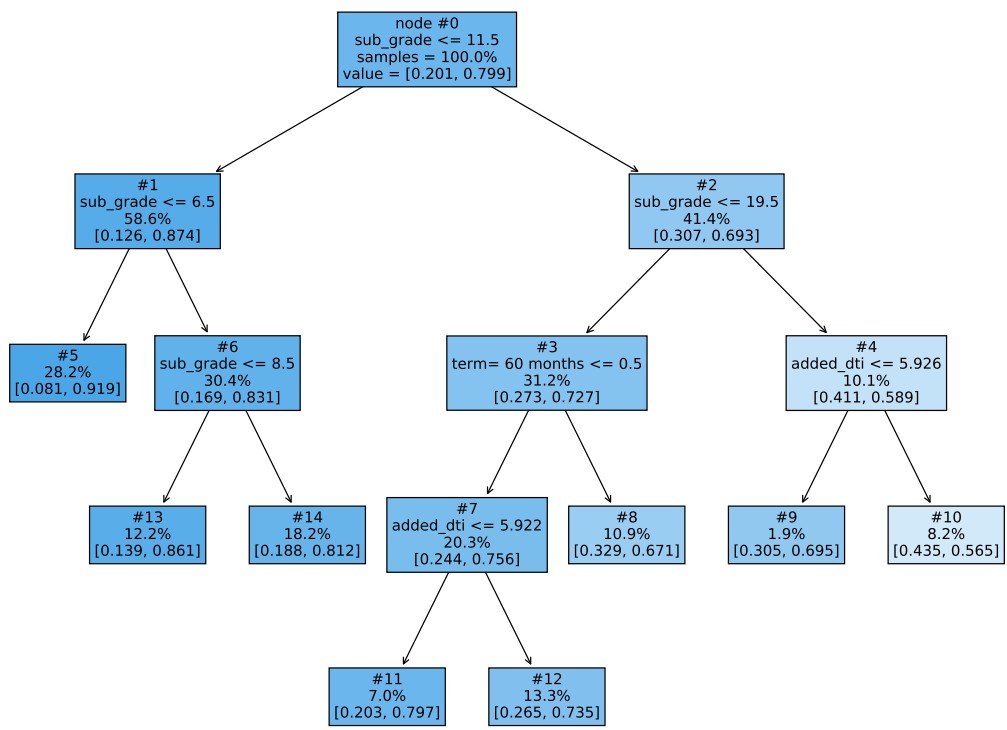

Figure 21: Decision tree reference model with 8 leaves for the Lending Club dataset.

**Reference model**    Figure 21 depicts the 8-leaf DT reference model for the Lending Club dataset. Most of the splits partition the 'sub_grade' feature, which is a measure of the quality of the loan (0–34 range, lower is better). Node 3 differentiates between 60-month terms and 36-month terms (the only alternative), while nodes 4 and 7 split on 'added_dti' (added debt-to-income ratio), which is the ratio between 12 months worth of the loan's payment installments and the borrower's annual income. While the structure of the DT agrees with domain knowledge (lower 'sub_grade' and lower 'added_dti' correlate with higher repayment probability), the test set accuracy of $79.8\%$ is no better than that of the trivial predictor that always returns the majority class of "paid off". The DT's AUC of $0.689$ however does indicate an improvement over the trivial predictor.

**LR and GAM models**    Tables 9 and 10 show the statistics of the LR and GAM classifiers that were trained on the Lending Club data. Similar to the DT reference model, the difference compared to Tables 5, 6 for the Adult Income dataset is that the accuracies remain no better than that of the trivial predictor, while the AUC does not show much increase either. These statistics suggest that the prediction task is difficult with the features available. Figures 22 and 23 display plots for the LR model with $C = 0.01$ and GAM with `max_bins = 8`, which are again chosen as representative models. The GAM in particular shows sensible monotonic behavior as functions of 'sub_grade', 'int_rate' (interest rate), 'dti' (debt-to-income ratio), etc., despite the unimpressive accuracy.

| $C$ | nonzeros | $\ell_1$ norm | accuracy | AUC |
|---|---|---|---|---|
| 1e-4 | 1 | 0.4 | 0.798 | 0.693 |
| 3e-4 | 2 | 0.6 | 0.799 | 0.695 |
| 1e-3 | 6 | 1.0 | 0.799 | 0.702 |
| 3e-3 | 8 | 1.3 | 0.799 | 0.702 |
| 1e-2 | 12 | 1.6 | 0.800 | 0.702 |
| 3e-2 | 17 | 2.1 | 0.799 | 0.703 |
| 1e-1 | 22 | 3.1 | 0.799 | 0.703 |
| 3e-1 | 24 | 3.7 | 0.799 | 0.703 |
| 1e+0 | 26 | 4.1 | 0.799 | 0.703 |
| 3e+0 | 26 | 4.2 | 0.799 | 0.703 |

Table 9: Number of nonzero coefficients, $\ell_1$ norm of coefficients, test set accuracy, and AUC for logistic regression models on the Lending Club dataset as a function of inverse $\ell_1$ penalty $C$.

| max_bins | accuracy | AUC |
|---|---|---|
| 4 | 0.798 | 0.696 |
| 8 | 0.798 | 0.703 |
| 16 | 0.799 | 0.704 |
| 32 | 0.799 | 0.705 |
| 64 | 0.799 | 0.705 |
| 128 | 0.799 | 0.705 |
| 256 | 0.799 | 0.705 |
| 512 | 0.799 | 0.705 |
| 1024 | 0.799 | 0.705 |

Table 10: Test set accuracy and AUC for Explainable Boosting Machines on the Lending Club dataset as a function of `max_bins` parameter.

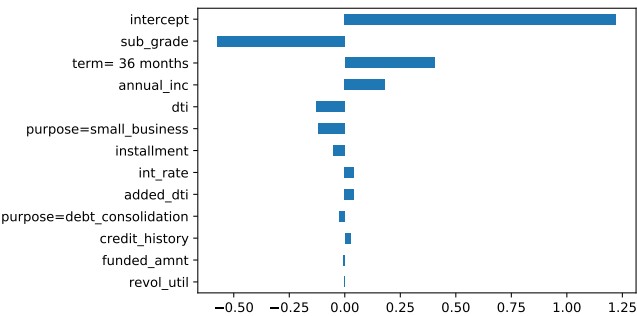

Figure 22: Coefficient values of the logistic regression model with $C = 0.01$ (12 nonzeros) for the Lending Club dataset.

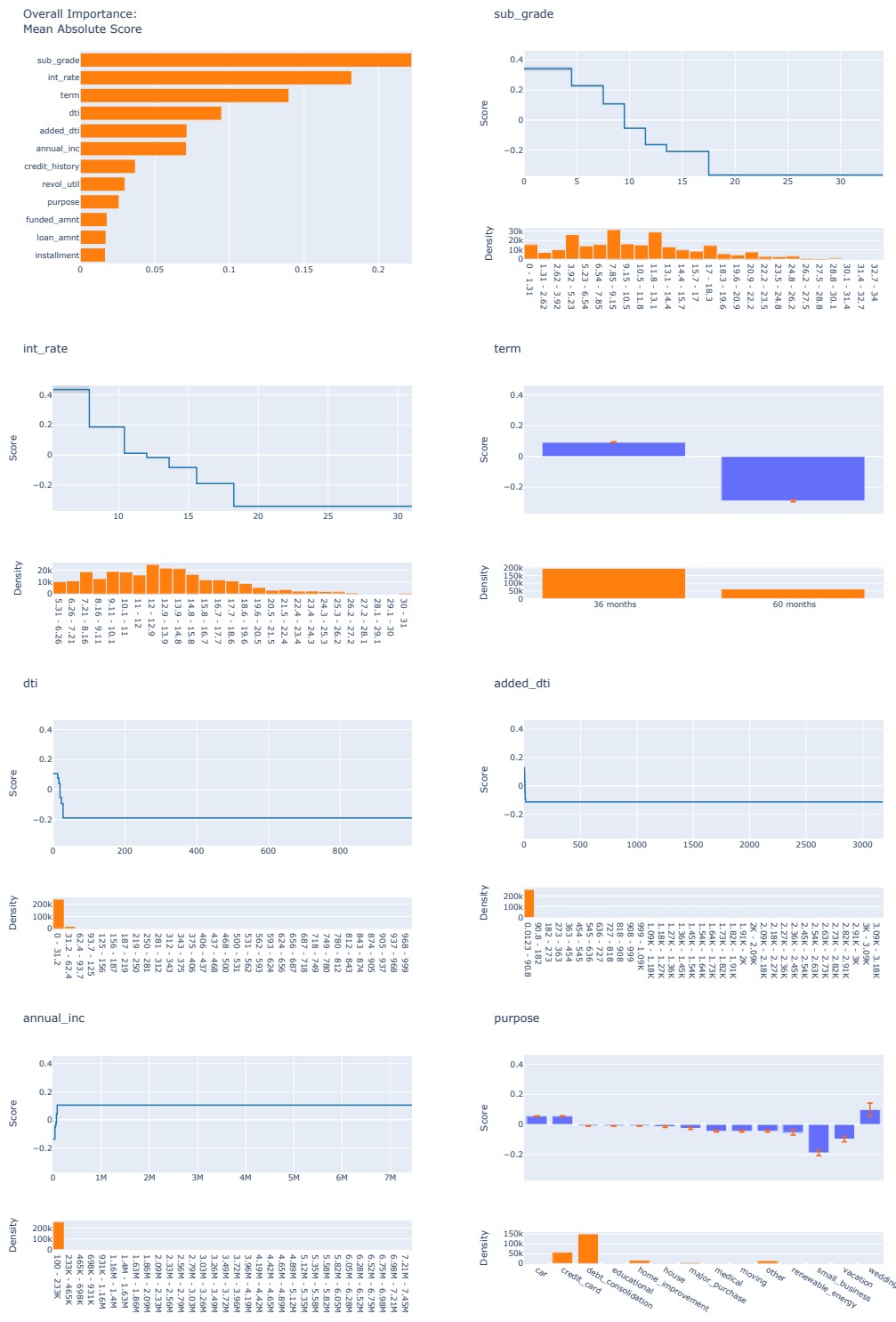

Figure 23: Feature importances and selected univariate functions $f_j$ for the Explainable Boosting Machine with `max_bins = 8` on the Lending Club dataset.

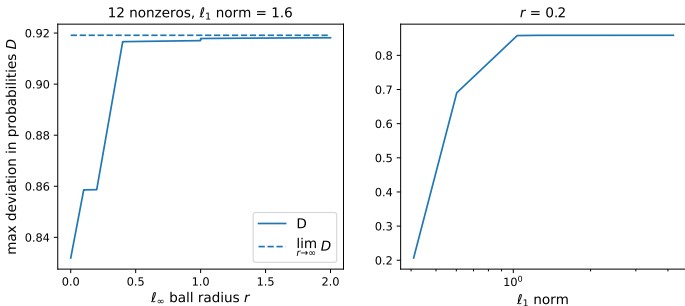

Figure 24: Maximum deviation $D$ for logistic regression models on the Lending Club dataset as a function of certification set size (radius $r$) and model smoothness ($\ell_1$ norm).

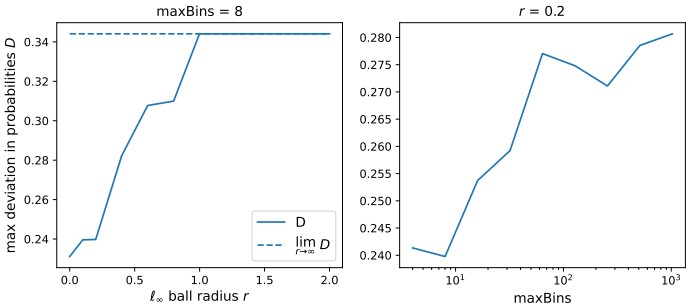

Figure 25: Maximum deviation $D$ for Explainable Boosting Machines on the Lending Club dataset as a function of certification set size (radius $r$) and model smoothness (`max_bins` parameter).

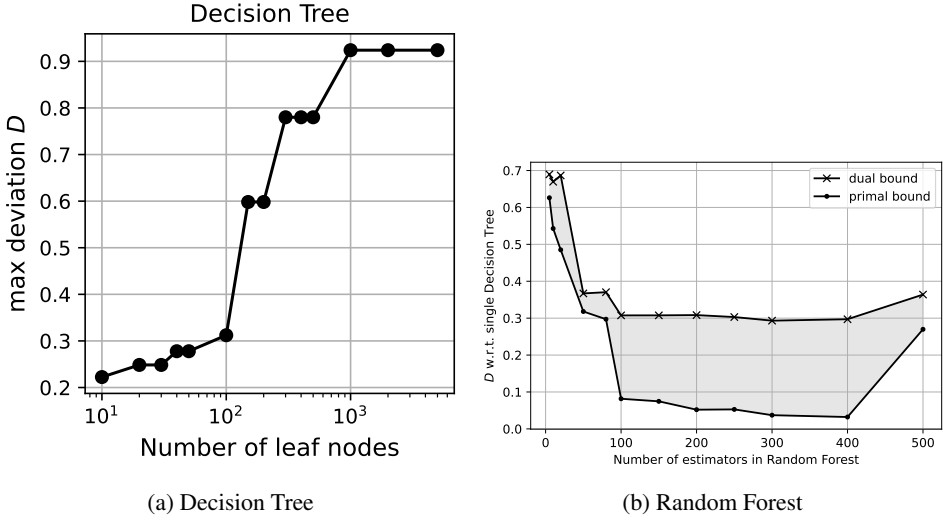

(a) Decision Tree                    (b) Random Forest

Figure 26: Maximum deviations computed for tree and tree ensembles on the Lending Club dataset

**Maximum deviation summary**   Figures 24–26 show maximum deviation as functions of certification set radius $r$ and model complexity parameters, in a similar manner as Figure 11 for the Adult Income dataset. The qualitative patterns are similar to before: increasing maximum deviation in all cases except with the number of RF estimators in Figure 26b, where the upper bound is stable around 0.7. A major quantitative difference is that the maximum deviations for the GAM in Figure 25 are much lower than for the other models, in particular LR in Figure 24. This is likely due to the fact that the GAM functions $f_j$ in Figure 23 are bounded while still being monotonic, unlike the linear functions $w_j x_j$ in the LR model.

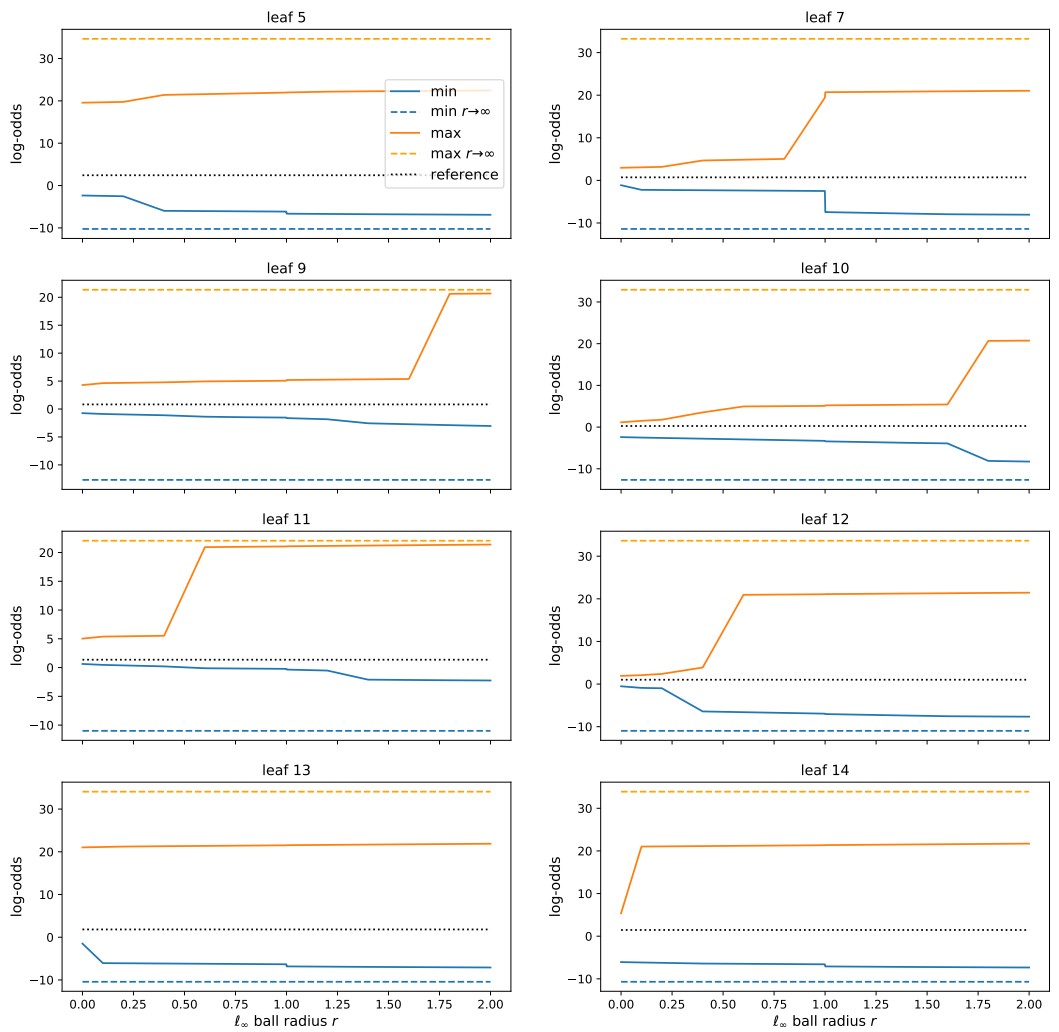

Figure 27: Minimum and maximum predicted log-odds for a logistic regression model with inverse $\ell_1$ penalty $C = 0.01$ on the Lending Club dataset, as a function of certification set size (radius $r$) and broken down by leaves of the decision tree reference model.

**Breakdown by leaves of** $f_0$     Figures 27–30 show a breakdown of the deviations for LR and GAM by leaves of the reference model, similar to Figures 15–18 and again on the log-odds scale. One difference is that in Figure 27, the deviations for finite $r$ do not come close to their $r \to \infty$ counterparts in most cases. In Figure 29 however, the $r \to \infty$ values are all attained when $r$ is slightly greater than $1$.

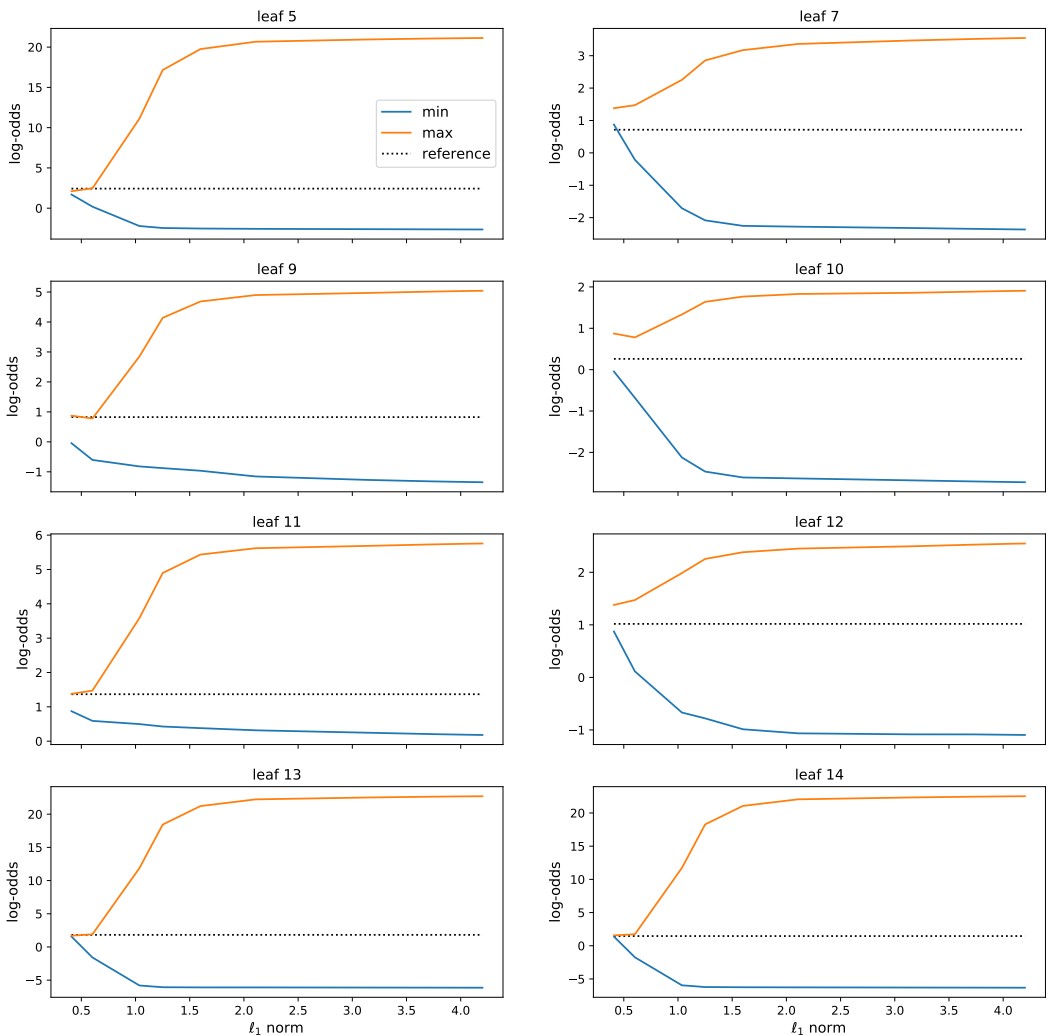

Figure 28: Minimum and maximum predicted log-odds for logistic regression models with different $\ell_1$ penalties $C$ on the Lending Club dataset, broken down by leaves of the decision tree reference model. The certification set $\ell_\infty$ ball radius is $r = 0.2$.

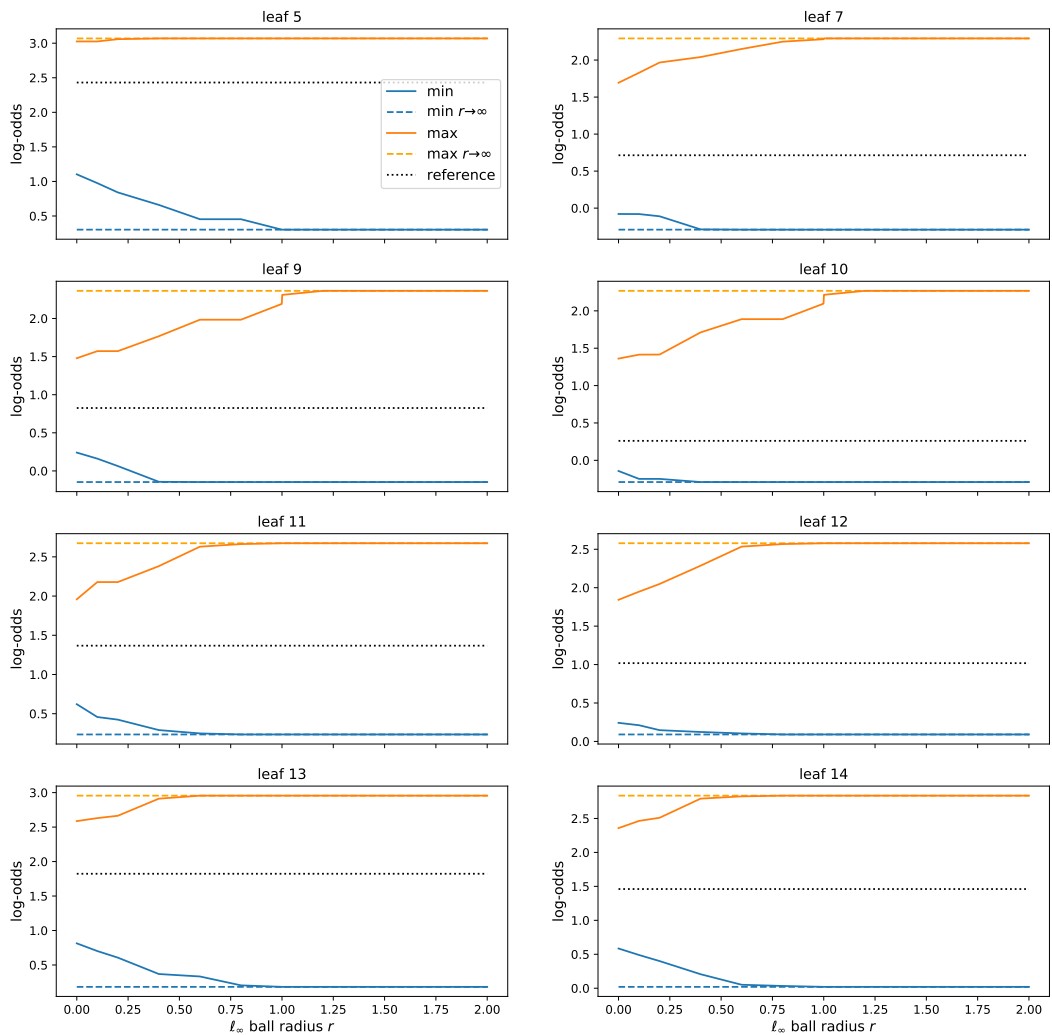

Figure 29: Minimum and maximum predicted log-odds for an Explainable Boosting Machine with `max_bins` = 8 on the Lending Club dataset, as a function of certification set size (radius $r$) and broken down by leaves of the decision tree reference model.

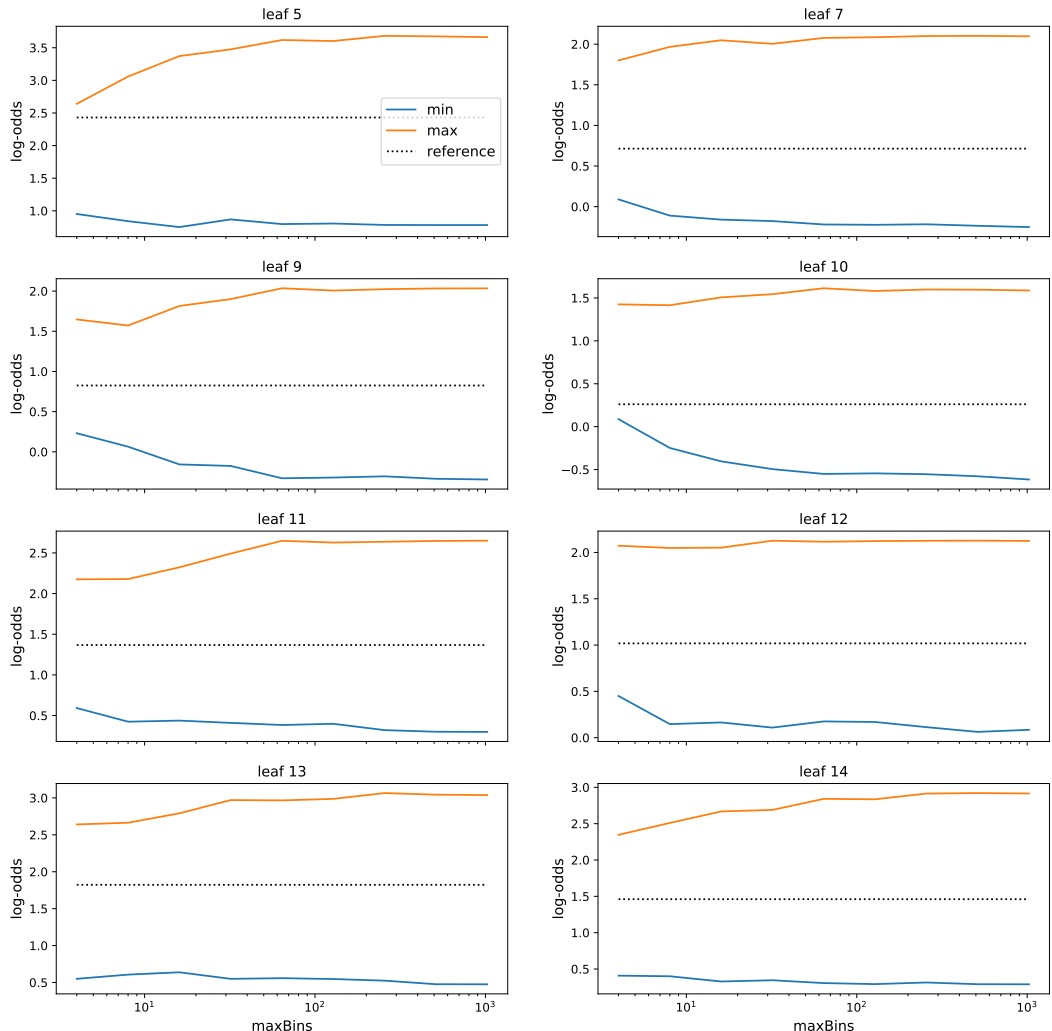

Figure 30: Minimum and maximum predicted log-odds for Explainable Boosting Machines with different `max_bins` values on the Lending Club dataset, broken down by leaves of the decision tree reference model. The certification set $\ell_\infty$ ball radius is $r = 0.2$.

**Running time, maximal cliques evaluated** Figure 31 shows the time required to compute the maximum deviation for LR and GAM on the Lending Club dataset. Figure 32 shows the number of $K + 1$-maximal cliques evaluated for DT and RF as well as the number of nodes in the graph. The observations are the same as in Figures 19 and 20.

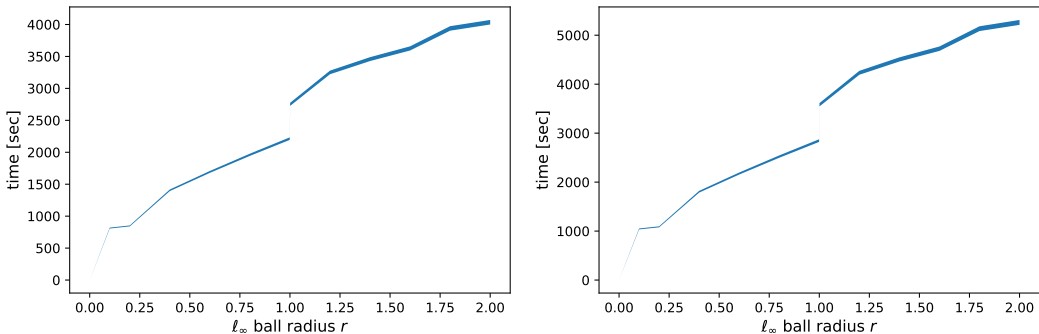

Figure 31: Time to compute maximum deviation for logistic regression models (left) and Explainable Boosting Machines (right) on the Lending Club dataset as a function of certification set size (radius $r$). The filled-in region shows the min-max variation with model complexity ($\ell_1$ norm for LR, `max_bins` for EBM).

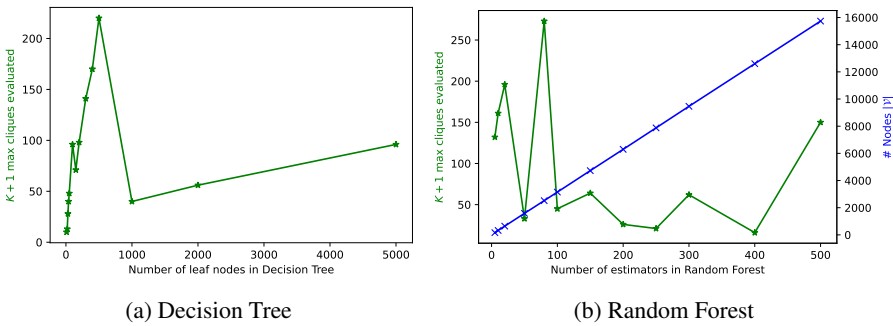

(a) Decision Tree                   (b) Random Forest

Figure 32: Effectiveness of pruning by bound for tree-based models on the Lending Club dataset

**Feature combinations that maximize deviation** Tables 11 and 12 present the feature values that maximize deviation for LR and GAM respectively. For both tables, the minimum log-odds over leaf 5 of the DT reference model is selected (corresponding to Figures 27 and 29, leaf 5, blue curve) because this choice maximizes the deviation overall for most values of $r$.

The previous trend toward extreme feature values that minimize log-odds also holds here as $r$ increases. For example, debt-to-income ratios increase to outlier values above $100\%$, annual income drops to the minimum of \$100, the term changes to 60 months in Table 11 for $r > 1$, and the purpose changes to small business, the category with the lowest log-odds in Figure 23. The decrease in income and increases in debt-to-income ratios are qualitatively in accordance with each other. However, these quantities are related to each other by deterministic formulas (at least in theory) that also involve the interest rate and installment amount. It is not clear whether the values in Tables 11 and 12 violate these relationships. This may be an instance that could benefit from constraints on possible feature combinations, as briefly mentioned in Appendix C.

| $r$ | dti | annual_inc | term | purpose | int_rate | credit_history |
|---|---|---|---|---|---|---|
| 0.0 | 505 | 1700 | 36 months | credit_card | 7.4 | 2557 |
| 0.1 | 506 | 100 | 36 months | credit_card | 6.9 | 2283 |
| 0.2 | 507 | 100 | 36 months | credit_card | 6.4 | 2009 |
| 0.4 | 884 | 100 | 36 months | debt_consolidation | 10.1 | 3897 |
| 0.6 | 886 | 100 | 36 months | debt_consolidation | 9.1 | 3349 |
| 0.8 | 889 | 100 | 36 months | debt_consolidation | 8.2 | 2801 |
| 0.999 | 891 | 100 | 36 months | debt_consolidation | 7.2 | 2256 |
| 1.001 | 891 | 100 | 60 months | small_business | 7.2 | 2251 |
| 1.2 | 893 | 100 | 60 months | small_business | 6.3 | 1706 |
| 1.4 | 895 | 100 | 60 months | small_business | 5.3 | 1158 |
| 1.6 | 898 | 100 | 60 months | small_business | 5.3 | 1095 |
| 1.8 | 900 | 100 | 60 months | small_business | 5.3 | 1095 |
| 2.0 | 902 | 100 | 60 months | small_business | 5.3 | 1095 |
| $\infty$ | 999 | 100 | 60 months | small_business | 5.3 | 1095 |

Table 11: Feature values that minimize log-odds for a logistic regression model ($C = 0.01$) over leaf 5 of the decision tree reference model for the Lending Club dataset. The 6 features that contribute most to the minimum are shown as a function of certification set radius $r$.

| $r$ | term | int_rate | dti | purpose | annual_inc | added_dti |
|---|---|---|---|---|---|---|
| 0.0 | 60 months | 9.9 | 46.4 | debt_consolidation | 35000 | 25.5 |
| 0.1 | 60 months | [10.4 11. ] | [28.6 30.9] | debt_consolidation | [31800 38001] | [10. 11.2] |
| 0.2 | 60 months | [12. 13.1] | [27.8 31.8] | debt_consolidation | [36600 38001] | [10. 11.7] |
| 0.4 | 60 months | [13.6 14.3] | [27.8 30.4] | small_business | [ 100 38001] | [12.7 15.3] |
| 0.6 | 60 months | [15.6 16.3] | [27.8 36.1] | small_business | [19801 38001] | [12.7 15. ] |
| 0.8 | 60 months | [15.6 16.4] | [27.8 28.4] | small_business | [14402 38001] | [12.7 15.7] |
| 0.999 | 60 months | [18.2 18.4] | [27.8 38.8] | small_business | [33069 38001] | [12.7 14.5] |
| 1.001 | 60 months | [18.2 18.8] | [27.8 35.3] | small_business | [ 935 38001] | [12.7 18.5] |
| 1.2 | 60 months | [18.2 20.2] | [27.8 33.3] | small_business | [ 7602 38001] | [12.7 18.1] |
| 1.4 | 60 months | [18.2 21.2] | [27.8 35.6] | small_business | [ 100 38001] | [12.7 20.4] |
| 1.6 | 60 months | [18.2 19.2] | [27.8 29.9] | small_business | [ 100 38001] | [12.7 24.5] |
| 1.8 | 60 months | [18.2 26.1] | [27.8 28.5] | small_business | [ 100 38001] | [12.7 24.4] |
| 2.0 | 60 months | [18.2 27.1] | [27.8 30.7] | small_business | [ 100 38001] | [12.7 26.6] |
| $\infty$ | 60 months | [18.2 31. ] | [ 27.8 999. ] | small_business | [ 100 38001] | [ 12.7 3179.3] |

Table 12: Feature values that minimize log-odds for an Explainable Boosting Machine (`max_bins` = 8) over leaf 5 of the decision tree reference model for the Lending Club dataset. The 6 features that contribute most to the minimum are shown as a function of certification set radius $r$. For $r > 0$, the minimizing values of continuous features form an interval because the corresponding functions $f_j$ are piecewise constant.