# OpenReview forum: "On the Safety of Interpretable Machine Learning: A Maximum Deviation Approach"
_NeurIPS.cc/2022/Conference — NeurIPS 2022 Accept_

### Official Review · Reviewer_yACw · 2022-07-09

**Rating:** 5
**Confidence:** 5
**Soundness:** 3 good
**Presentation:** 3 good
**Contribution:** 2 fair

**Summary:**

This paper proposes to use maximum deviation, defined by the difference between a reference model and a trained model over a superset of the training data,  achieving safety and interpretability for supervised learning models.  Such a measure can be efficiently computed for decision trees, generalized linear and additive models. The idea of using maximum deviation to measure the model safety is novel.  However, the proposed approach is somewhat too restrictive to generalize.

**Questions:**

1. How to specify a reference model f_0 in practice? Since  we do not have any prior on the relationship between input and output,  an improper specification on f_0 may result in misleading interpretable result.

2. The theoretical part is not surprise to me. In particular, the authors argue that when f and f_0 are both decision trees, the complexity of computing maximum deviation cost  in O(\epsilon), which is actually stated in [1] for verifying the robustness of single tree. Besides, a pioneering work [2] proposed a verification of trees using anytime searching algorithm to address multiple verification tasks on ensemble trees. So the method described in section 4.3 is basically a generalization of [2].

3. Consider that modern machine learning models are usually with complex structure (e.g. deep neural networks) when dealing with big data. The proposed framework is too restrict to generalize. My concern lies with when the reference model f_0 is a complex DNN, is there an efficient way to compute the corresponding maximum deviation, especially when f is also a DNN. How about the corresponding computational complexity.

4. It would be better to add some baseline comparisons to illustrate the safety of a specific model.

[1] Hongge Chen, Huan Zhang, Si Si, Yang Li, Duane Boning, Cho-Jui Hsieh, robustness verification of tree-based models, NeurIPS, 2019.
[2] L. Devos, W. Meert and J. Davis, Versatile verification of tree ensembles, ICML, 2021.



**Limitations:**

Please refer to Questions

**Strengths And Weaknesses:**

Strengths:
1. The topic is quite interesting and important.
2. Rich experimental results.
3. Well written and organized.

---

> ### Author Response · Authors · 2022-08-02
> **Response to Reviewer yACw**
>
> Thank you very much for your time and valuable feedback! Please find below our responses to your questions. In particular, in Appendix B.5, we had already outlined possible approaches to generalize deviation maximization to neural networks in future work.
>
> > How to specify a reference model f_0 in practice? Since we do not have any prior on the relationship between input and output, an improper specification on f_0 may result in misleading interpretable result.
>
> Please see our response to all reviewers under "Choice of reference model" and "Reference model may be incorrect/improper."
>
> > The theoretical part is not surprise to me. In particular, the authors argue that when f and f_0 are both decision trees, the complexity of computing maximum deviation cost in O(\epsilon), which is actually stated in [1] for verifying the robustness of single tree. Besides, a pioneering work [2] proposed a verification of trees using anytime searching algorithm to address multiple verification tasks on ensemble trees. So the method described in section 4.3 is basically a generalization of [2].
>
> For the case where $f$ and $f_0$ are decision trees, we now cite Theorem 1 of Chen et al. [1] (in the case of a $K=2$-tree ensemble) as a result related to Proposition 1, although our Proposition 1 is slightly more precise in that $f$ and $f_0$ do not have to have the same numbers of leaves $L$ and $L_0$, and we exactly enumerate the edges, $|\mathcal{E}| \leq L_0 L$. We agree with the other points and had already acknowledged them in the paper: Section 4.3 gives credit to Devos et al. [2], while we acknowledge in line 146 that some results are less surprising. As we state there, we think our contribution is to identify precise properties that allow the results to hold. We also kindly remind the reviewer of our other contributions, namely the framework of deviation maximization and the case studies (Section 5 and the appendix) showing that it can provide insights.
>
> > My concern lies with when the reference model f_0 is a complex DNN, is there an efficient way to compute the corresponding maximum deviation, especially when f is also a DNN.
>
> Please see our response to all reviewers under "Deep neural networks."
>
> > It would be better to add some baseline comparisons to illustrate the safety of a specific model.
>
> We are not sure what you mean by this. If you mean comparing our methods to baselines in the case studies in Section 5, we are not aware of such baselines since we believe that the conceptual framework of maximum deviation is new.

---

> > ### Comment · Reviewer_yACw · 2022-08-05
> > **Response to response**
> >
> > Thanks for the authors' response. Most of my concern are well addressed.  I would like to raise my score to Borderline accept.

---

### Official Review · Reviewer_V5qC · 2022-07-12

**Rating:** 5
**Confidence:** 1
**Soundness:** 3 good
**Presentation:** 2 fair
**Contribution:** 2 fair

**Summary:**

This paper defines a new "safety" measure for a ML model given a good reference model (normally an interpretable model). The safety metric is the maximum of outputs deviation between the target model and the reference model for some input sets. The authors then analyze the "safety" metric for some common ML models, including decision trees, generalized linear and additive models, tree ensembles, and piecewise Lipschitz functions. The authors then show how "safety" metric on two real world datasets.

**Questions:**

- An interpretable model may sound good, but it won't get 100% correctness. Often interpretable model will sacrifice performance for interpretability. What if the baseline model is wrong on some inputs but the target model is actually correct. In that case, how would the safety measure address this issue?
- In practice, how do you choose the certification set? You may not have access to the test set in reality and domain experts' inputs may just be a few data samples with label. In that case, how would one define the certification set based on the limited inputs?
- This question might be less related. But I'm wondering, how would this safety metric be used by real-world ML applications? Do the users train a few models and choose the safest one by the metric regardless of the model's overview performance? If the certification set contains a few data samples, can the user just run a sanity check for each model and drop those models with unsafe outputs?

**Ethics Review Area:**

["I don’t know"]

**Limitations:**

- I think this paper describes how to compute the safety score for some sets of common ML models. Extra work is required to generalize this metric to all ML models.
- Depends on the certification set, it may not be practical to compute the safety score for some ML models.

**Strengths And Weaknesses:**

Strengths:
- The idea of using a reference model to measure how safe another model is looks quite novel.
- The authors describe the motivation and the actual idea well. Most parts of this paper are relatively easy to follow.

Weaknesses:
- The discussions on how the baseline model and the certification set are chosen are quite limited. It was not very convincing that the current form is what the idea of two model safety measure would be. See my questions in the section below.
- The experimental section is not easy to follow. There are too many discussions/figures or other information on the appendix. Readers have to go back and forth multiple times between the main body and appendix.

---

> ### Author Response · Authors · 2022-08-02
> **Response to Reviewer V5qC**
>
> Thank you very much for your time and valuable feedback! We hope that the following responses to your comments will bring greater clarity.
>
> > The discussions on how the baseline model and the certification set are chosen are quite limited.
>
> Please see our response to all reviewers under "Choice of reference model" and "Choice of certification set."
>
> > An interpretable model may sound good, but it won't get 100% correctness. Often interpretable model will sacrifice performance for interpretability. What if the baseline model is wrong on some inputs but the target model is actually correct. In that case, how would the safety measure address this issue?
>
> Please see our response to all reviewers under "Reference model may be incorrect/improper."
>
> > In practice, how do you choose the certification set? You may not have access to the test set in reality and domain experts' inputs may just be a few data samples with label.
>
> Please see our response to all reviewers under "Choice of certification set." Also just to be clear, by "domain expert input," we meant higher-level feedback along the lines of "these predictions in this input region make sense/do not make sense because ...," not labelling specific unlabelled samples.
>
> > How would this safety metric be used by real-world ML applications? Do the users train a few models and choose the safest one by the metric regardless of the model's overview performance? If the certification set contains a few data samples, can the user just run a sanity check for each model and drop those models with unsafe outputs?
>
> We hope actually that the maximum deviation will not be used as just another metric to be optimized. A small maximum deviation implies that the assessed model $f$ is close to the reference model everywhere and thus similarly safe. But as we write in lines 69-74, a large maximum deviation does not necessarily indicate a safety risk, but rather a direction to investigate further. This spirit is what we hoped to convey with the case studies in Section 5, which look into the feature values that maximize deviation.
>
> > The experimental section is not easy to follow. There are too many discussions/figures or other information on the appendix. Readers have to go back and forth multiple times between the main body and appendix.
>
> We tried our best under the page constraints imposed by NeurIPS. If the reviewer has specific suggestions, we would gladly take them into consideration.
>
> > I think this paper describes how to compute the safety score for some sets of common ML models. Extra work is required to generalize this metric to all ML models.
>
> We agree, and we hope that the suggestions in Appendix B.5 for neural networks and post hoc explanations will be useful.
>
> > Depends on the certification set, it may not be practical to compute the safety score for some ML models.
>
> You may be interested in our response to Reviewer aVuR on relaxing the Cartesian product assumption on $\mathcal{C}$.

---

### Official Review · Reviewer_aVuR · 2022-07-12

**Rating:** 6
**Confidence:** 4
**Soundness:** 3 good
**Presentation:** 3 good
**Contribution:** 3 good

**Summary:**

This paper proposes a new conceptual framework to assess the safety of model via maximum deviation from a reference model over a pre-specified certification dataset. The paper also motivates how such reference model along with certification dataset might be chosen depending on the problem. With this quantification of safety, the paper then demonstrates the computation of this deviation for interpretable models like trees and generalized additive models. They also discuss discrete optimization techniques to compute this deviation metric in case of tree ensembles which are not considered interpretable. Finally they discuss the case of a more general piecewise Lipschitz functions and show how interpretable models can lead to more efficient computation of their metric.

Through cases studies, the paper demonstrates how this framework can be used to identify regions within feature space where the deviation is large. This can be further subjected to investigation, helping in improving safety of models. They also demonstrate the case where deviation bounds may be used to study the effect of number of estimators in the random forest on safety of the model as defined within their framework.

**Questions:**

1) It would be interesting to have a set of thumbrules/guidelines on the selection of deviation function D(y, y_0) under different scenarios. Have the authors generated any results on this?
2) I would like to know if the authors have tried to assess models like deep neural networks (since they are not intepretable according to the assumptions in the paper - will this reduce to worst case scenario when computing maximum deviation?).



**Limitations:**

Yes, addressed in the appendix.

**Strengths And Weaknesses:**

Strengths:
1) The conceptual framework proposed in this paper is intuitive and helps reason about the safety of a predictive model.
2) The paper demonstrates the computation of deviation or bounds on deviation for trees, GAMs and tree ensembles along with a broader class of piecewise Lipschitz functions. This helps in understanding the practicality of this framework under different situations
3) The case studies show how this framework can be useful not only in the assessment of regions within feature space that are problematic, but also in choosing hyperparameters such that the models are safe w.r.t this deviation metric

Weaknesses:
1) Choosing a reference model, certification dataset and the deviation function D(y, y_0) may not be a trivial exercise while assessing the safety of non-interpretable model. It is not straightforward to reason about the tightness and usefulness of bounds computed under these circumstances.
2) In the case where the model is interpretable, we still need to make several assumptions (e.g. C is cartesian product while optimizing for non-linear additive f). While many of these assumptions made in the paper look reasonable, it may limit the applicability of the method to several practical scenarios involving interpretable models.

---

> ### Author Response · Authors · 2022-08-02
> **Response to Reviewer aVuR**
>
> Thank you very much for your time and valuable feedback! We hope that the following responses to your points will bring greater clarity.
>
> > Choosing a reference model, certification dataset and the deviation function D(y, y_0) may not be a trivial exercise
>
> Please see our response to all reviewers under "Choice of reference model" and "Choice of certification set." Our response regarding the deviation function is below.
>
> > In the case where the model is interpretable, we still need to make several assumptions (e.g. C is cartesian product while optimizing for non-linear additive f).
>
> As you probably recognize, we need some assumptions to make the deviation maximization in (1) tractable. For the specific example you cite, if we assume only that $f$ is additive but the shape functions $f_j$ can be general nonlinear, then the Cartesian product assumption greatly simplifies the problem by reducing it to multiple one-dimensional optimizations. If $\mathcal{C}$ is not a Cartesian product, then one way to still bound the maximum deviation is to find the smallest Cartesian product $\overline{\mathcal{C}}$ that contains $\mathcal{C}$ and maximize deviation over $\overline{\mathcal{C}}$. As long as it is relatively easy to optimize linear functions over $\mathcal{C}$, then constructing such a Cartesian product is similarly easy. Another conceivable relaxation of the Cartesian product assumption is a Cartesian product of low-dimensional sets, not just one-dimensional. We have added these possible relaxations to Appendix B.2.
>
> > It would be interesting to have a set of thumbrules/guidelines on the selection of deviation function D(y, y_0) under different scenarios.
>
> We have added the following discussion to Appendix A: For the case where $y$, $y_0$ are real-valued scalars (which covers binary classification and regression), while we stated Assumption 1 as a sufficient condition for tractable optimization with GAMs, we think that it is also an intuitively reasonable requirement: the deviation should increase the farther $y$ is from $y_0$ in either direction. In addition, symmetry may be desirable, i.e. $D(y, y_0) = D(y_0, y)$, to not favor one of the two models over the other. Both Assumptions 1 and 2 as well as symmetry are satisfied by monotonically increasing functions $D(|y - y_0|)$ of the absolute difference, for example powers $|y - y_0|^p$ for $p > 0$. For the case where $y, y_0 \in \mathbb{R}^M$ as in multi-class classification, we think that it is advantageous for $D(y, y_0)$ to decompose into a sum over output dimensions: $D(y, y_0) = \sum_{k=1}^M D_k(y_k, y_{0k})$. For example, the ($p$th power of the) $\ell_p$ distance $\lVert y - y_0 \rVert_p^p = \sum_{k=1}^M |y_k - y_{0k}|^p$ is separable in this manner.
>
> > I would like to know if the authors have tried to assess models like deep neural networks (since they are not intepretable according to the assumptions in the paper - will this reduce to worst case scenario when computing maximum deviation?).
>
> Please see our response to all reviewers under "Deep neural networks." In short, by leveraging the approaches that we suggest, it should not come down to a "worst case scenario" (by which we assume you mean exhaustive evaluation).

---

> > ### Comment · Reviewer_aVuR · 2022-08-07
> > **Thank you for your response**
> >
> > Thank you, this answers my questions. I would like to keep my score unchanged.

---

### Author Response · Authors · 2022-08-02
**Response to All Reviewers**

Thanks to all the reviewers for your efforts. We are encouraged that you find the concept of maximum deviation to be "quite novel" and intuitive, the case studies to be useful and rich, and the writing to be organized and easy to follow. Below we respond to common issues raised in your reviews. We then respond to the remaining comments individually. Changes to the paper are highlighted in blue in the updated version.

### **Choice of reference model (all reviewers)**

We realize that the discussion in the main paper (Section 2, Reference Model paragraph) was indeed brief due to the page limit. In our reply to this comment, we expand on the examples given in the Reference Model paragraph to give a better sense of how a reference model could be chosen. These examples can be categorized as 1) existing domain-specific models, 2) interpretable ML models validated by domain knowledge, and 3) extensively tested and deployed models. We have also added this expanded discussion to Appendix A in the revised manuscript. If our paper is accepted, this expanded discussion will be moved into the main paper as part of the additional page allowed in the camera-ready version (specifically, it will replace lines 86-92 in the current manuscript).

### **Reference model may be incorrect/"improper" (Reviewers V5qC, yACw)**

As we wrote in Section 2, Reference Model paragraph, it is indeed more important for the reference model to represent expected behavior, either according to domain expertise or because it has been extensively tested, than to be the most accurate, although higher accuracy is of course desirable. The proposed safety assessment can in fact provide a way of correcting the reference model, identifying where the expectations encoded in it may be incorrect or too simplistic (i.e., the assessment can work in both directions). The "deviation can be good" example in Section 5 is an example of the assessed model $f$ providing more informative predictions than the reference model. We have added a note to this effect in the revised manuscript.

### **Deep neural networks (Reviewers yACw, aVuR)**

1. First, as we note in lines 48-49, 151-152, the black-box method of Section 4.4 applies to NNs, but will clearly be less efficient than a method that is aware of the NN structure.
2. As for a NN-specific method, we believe that the present state of ML research offers some plausible avenues toward this, and thus disagree with Reviewer yACw that "the proposed framework is too restrictive to generalize." At the same time, doing so would be well beyond the scope of a single paper and possibly another paper in itself. In Appendix B.5, we had already suggested possible approaches starting from the literature on NN robustness verification and post hoc explanation. Please see Appendix B.5 as well as our reply to this comment for specifics.

### **Choice of certification set (Reviewers V5qC, aVuR)**

We do not think that the choice of dataset is critical to specifying the certification set as it depends only weakly on the dataset. Specifically, the dependence is only on an expanded version of the support of the dataset and not on other aspects of the data distribution (see lines 96-97). In the case of the union of balls (2), for larger radii $r$ (and hence more comprehensive $\mathcal{C}$), the balls overlap significantly and the exact locations of the ball centers (the data points) become less important. Ideally, the certification set should not depend on a dataset at all since it is intended to capture out-of-distribution risks. In practice however, some data is needed to determine the input domain and the data manifold in high-dimensional space.
- For Reviewer V5qC: In light of the above, lack of access to the test set does not matter (we have noted this in lines 323-324). We just need some dataset that furthermore does not have to be labelled.

---

> ### Author Response · Authors · 2022-08-02
> **Categories and examples of reference models**
>
> 1. **Existing domain-specific models:** In consumer finance, several industry-standard models compute credit scores from a consumer's credit information (the FICO score is the best-known in the US). Similarly in medicine, scoring systems (sparse linear models with small integer coefficients) abound for assessing various risks (the CHADS$_2$ score for stroke risk is well-known, see the "Scoring Systems: Applications and Prior Art" section of [Rudin and Ustun (2018)](https://users.cs.duke.edu/~cynthia/docs/WagnerPrizeJournal.pdf) for a list of others). These models have been used for decades by thousands of practitioners so they are very well understood. They may very well be improved upon by a more ML-based model, but for such a model to gain acceptance with domain experts, any large deviations from existing models need to be examined and understood.
> 1. **Interpretable models validated by domain knowledge:** Here, an interpretable ML model is learned from data and is validated by domain experts in some way, for example by selecting important input features or by carefully inspecting the trained model. We provide two real examples:
>     - In semiconductor manufacturing, process engineers typically want decision trees ([Dhurandhar et al., NeurIPS 2018](https://proceedings.neurips.cc/paper/2018/hash/972cda1e62b72640cb7ac702714a115f-Abstract.html)) to model their respective manufacturing process (e.g. Etching, Polishing, Rapid Thermal Processing, etc.) since they are comfortable understanding and explaining them to their superiors, which is critical especially when things go out-of-spec. Hence, a tree built from data (or any model in general) would only be allowed to make automated measurement predictions if the features it highlights (viz. pressures, gas flows, temperatures) make sense for the specific process.
>     - Similarly, in predicting failures of industrial assets such as wind turbines, some failure data is available to train models but we may also consult with experts in these systems (e.g. engineers). They have knowledge that can help validate the model, for example which components are most likely to cause failures or which environmental variables (e.g. temperature) are most influential.
> 1. **Extensively tested and deployed models:** A reference model may also be one that is not necessarily informed by domain knowledge but has been extensively tested, deployed, and/or approved by a regulator.
>     - For medical devices that use ML models, the US Food and Drug Administration (FDA) has instituted a risk-based regulatory system. Any system updates or changes, for instance changes in model architecture, retraining based on new data, or changes in intended use (e.g. use for pediatric cases for devices approved only for adults), need to either seek new approvals or demonstrate "substantial equivalence" by providing supporting evidence that the revised model is similar to a previously approved device. In the latter case, the reference model is the approved device and small maximum deviation serves as evidence of equivalence.
>     - As another example, consider a ML-based recommendation model for products of an online retailer or articles on a social network, where because of the scale (serving billions of users), a tree ensemble may be used for its fast inference time as well as its modeling flexibility (see this [blog post](https://engineering.fb.com/2017/03/27/ml-applications/evaluating-boosted-decision-trees-for-billions-of-users/)). In this case, a (tree ensemble) model that has been deployed for some time could be the reference model, since it has been extensively tested during this time even though human validation of it may be limited. When a new version of the model is trained on newer data or improved in some fashion, finding its maximum deviation from the reference model can serve as one safety check before deploying it in place of the reference model.

---

> > ### Author Response · Authors · 2022-08-02
> > **More on neural networks (already in Appendix B.5)**
> >
> > - Appendix B.5 discusses how the problem of robustness verification for NNs, which is a non-trivial problem in itself with an extensive literature, is essentially a single-model case of our problem (1) in which $f_0$ is a constant (and with an appropriate choice of the deviation function $D(f, f_0)$). We would expect that solutions to a two-NN verification problem would leverage existing robustness verification methods, but developing such solutions entails substantial work. In addition, existing studies of verification methods have largely focused on local neighborhoods (with radii $\epsilon \leq 0.1$ in terms of normalized feature values) and further investigation is needed to see how these methods fare for larger radii in our context.
> >
> > - Another possibility for NNs, as discussed in Appendix B.5, is to make use of post hoc explanations in the form of a simpler proxy model $\hat{f}$. Here we identify the problem to be one of providing post hoc explanations with *uniform* guarantees on fidelity (over a local or global region). We are not aware of existing post hoc explanations that provide such a guarantee, and we mention a possible approach using quantile regression. Assuming such uniform proxies can be constructed, then for certain modalities or applications it may be possible to train highly accurate proxies. For instance for tabular data, random forests or boosted trees might very well replicate the behavior of a NN, in which case the machinery introduced in Section 4.3 could be readily used. Even for other modalities such as text and images, interpretable models such as [Neural Additive Models (NAMs)](https://papers.nips.cc/paper/2021/hash/251bd0442dfcc53b5a761e050f8022b8-Abstract.html) and [continued fraction networks (CoFrNets)](https://papers.nips.cc/paper/2021/hash/b538f279cb2ca36268b23f557a831508-Abstract.html) may prove to be sufficient in some cases.
> >
> > - There are recent architectures such as Lipschitz NN ([Meunier et al., ICML 2022](https://proceedings.mlr.press/v162/meunier22a.html)) which are adversarially robust and hence valuable in practice. Our analysis presented in Section 4.4 for piecewise Lipschitz models would be applicable here, where the simple regret of standard bandit algorithms for a given number of queries could be reduced to equation 14 (as opposed to equation 13) in the paper.

---

### Author Response · Authors · 2022-08-05
**Any further questions or clarifications?**

A gentle reminder to the reviewers that if you have any further questions for us or clarifications of your comments, please post them soon so that we have time to respond to them before the end of the author-reviewer discussion next Tuesday. Thanks!

---

### Public Comment · Authors · 2022-12-16
**full version**

Please see the supplementary material or https://arxiv.org/abs/2211.01498 for the full version with working hyperlinks.

---

### Meta-Review · Area_Chair_UMhA · 2022-08-26

**Recommendation:** Accept
**Confidence:** Less certain

**Metareview:**

The authors propose to inspect learned models based on their maximum deviation to a reference model. They evaluate the feasibility of computing this deviation for a number of widely used model classes, including generalised linear models and decision trees. The idea is illustrated in cases studies.

Reviewers all appreciated the novelty and importance of the proposed problem and the contributions made to examine the feasibility of solving it. Their main concerns were regarding the limited discussion on choosing the reference model and the certification set. The authors expanded greatly on this in their rebuttal and in a revision to the paper.

The technical novelty is rather lower but the paper should not have trouble finding an audience in the NeurIPS community due to the broad applicability of the problem under consideration the clarity of the manuscript. I think the added discussion asked for by reviewers is well within what could be expected to be added for a camera-ready version (and indeed, this is already in the Appendix of the revision).

**Award:**

No

---

### Decision · Program_Chairs · 2022-09-14

Accept